# JOINTLY: interpretable joint clustering of single-cell transcriptomes

Andreas Fønss Møller [1,2] & Jesper Grud Skat Madsen [1,3,4,5] ✉

Single-cell and single-nucleus RNA-sequencing (sxRNA-seq) is increasingly being used to characterise the transcriptomic state of cell types at homeostasis, during development and in disease. However, this is a challenging task, as biological effects can be masked by technical variation. Here, we present JOINTLY, an algorithm enabling joint clustering of sxRNA-seq datasets across batches. JOINTLY performs on par or better than state-of-the-art batch integration methods in clustering tasks and outperforms other intrinsically interpretable methods. We demonstrate that JOINTLY is robust against overcorrection while retaining subtle cell state differences between biological conditions and highlight how the interpretation of JOINTLY can be used to annotate cell types and identify active signalling programs across cell types and pseudo-time. Finally, we use JOINTLY to construct a reference atlas of white adipose tissue (WATLAS), an expandable and comprehensive community resource, in which we describe four adipocyte subpopulations and map compositional changes in obesity and between depots.

Single-cell and single-nucleus RNA-sequencing (sxRNA-seq) has immense potential to enhance our understanding of human biology during homeostasis and how development or diseases shape our cells, tissues, and organs. To relate gene expression programs in specific cell types or cell states to a disease or developmental state, it is required that batch effects, which are technical sources of variation, between samples are removed, as they can otherwise introduce false or mask true associations.

In recent years, several methods have attempted to overcome the problem of batch effects by integrating sxRNA-seq datasets using graph-based (e.g., fastMNN[1]), statistical (e.g., Harmony[2]), or deep learning-based (e.g., scVI[3]) approaches. However, it remains a challenging problem. A recent comprehensive benchmark of batch integration methods by the Theis group found that each method has a different balance between conserving biological variation and removing batch effects and that this balance can depend on the integration task[4]. In addition to conserving different amounts of biological variation, each method also has different and task-specific sensitivity to over-correction, where biological variation, rather than batch

effects, are removed[5,6]. Interpretable batch integration methods can aid in evaluating the integration performance in a specific task by enabling the user to evaluate whether the genes or gene modules, driving integration are meaningful in the biological context.

There are already interpretable batch integration methods, such as LIGER[7], which is based on non-negative matrix factorisation-based (NMF). LIGER learns shared (or dataset-specific) factors that in linear combination can describe each cell, and these factors are interpretable as they are defined by non-negative linear combinations of genes. Thus, the factors, that can be used for clustering, also represent weighted gene modules that allow the user to evaluate the integration and can assist in functional annotation of datasets. However, in benchmarks, such linear methods are not efficient at removing batch effects[4] likely because batch effects can be highly non-linear[8].

Here, we introduce JOINTLY, a hybrid linear and non-linear NMF-based joint clustering tool. We benchmark JOINTLY and eight other batch integration methods in five different integration tasks composed of a total of 52 datasets. JOINTLY achieves state-of-the-art performance and additionally generates interpretable factors.

[1]Institute of Biochemistry and Molecular Biology, University of Southern, Odense, Denmark. [2]Sino-Danish College (SDC), University of Chinese Academy of Sciences, Beijing, China. [3]Institute of Mathematics and Computer Science, University of Southern Denmark, Odense, Denmark. [4]Center for Functional Genomics and Tissue Plasticity (ATLAS), Odense M 5230, Denmark. [5]The Novo Nordisk Foundation Center for Genomic Mechanisms of Disease, Broad Institute of MIT and Harvard, Cambridge, MA 02142, USA. ✉e-mail: jgsm@imada.sdu.dk

We show that the genes associated with the interpretable factors are more specific for cell types than traditional marker genes and that they can be used to guide cell type annotation and discover active biological processes across cell types and pseudo-time. We evaluate the robustness against over-correction by integrating multi-donor datasets from different tissues and find that JOINTLY removes within-tissue batch effects but retains across-tissue biological variability. Finally, we demonstrate how JOINTLY can be used to create a tissue atlas by clustering and labelling cell types and states in white adipose tissue from six different studies. Based on these high-quality labels, we create a reference atlas of white adipose tissue (WATLAS) deeply characterising the transcriptome of 43 cell types and states. The WATLAS is a community resource, which is a source for hypothesis generation, for contextualising new datasets through co-embedding and cell type and state annotation using transfer learning, as well as for use as a reference for deconvolution. Analysis of WATLAS revealed compositional differences between lean and obese donors and between different white adipose tissue

depots, which we support by deconvoluting bulk RNA-sequencing samples from approximately 1300 additional donors.

## Results

### JOINTLY identifies joint clusters and shared gene modules

There are at least two major sources of variation in multi-sample single-cell and single-nucleus RNA-seq (sxRNA-seq), namely biological signals and batch effects. The objective of our method, named JOINTLY, is to exclusively capture the biological signals for cell clustering (Fig. 1A). To achieve this, JOINTLY employs a hybrid framework of linear and non-linear non-negative matrix factorisation. This framework optimises three distinct low-rank matrices through multiplicative updating. The non-linear component of JOINTLY decomposes the gene expression matrix using consensus PCA (see "Methods") into a reduced dimensional space that effectively captures the shared variance between batches, while also accounting for residual dataset-specific variance. For each dataset, this reduced dimensional space is used to estimate local cell–cell distances using an adaptive heat-based kernel[9].

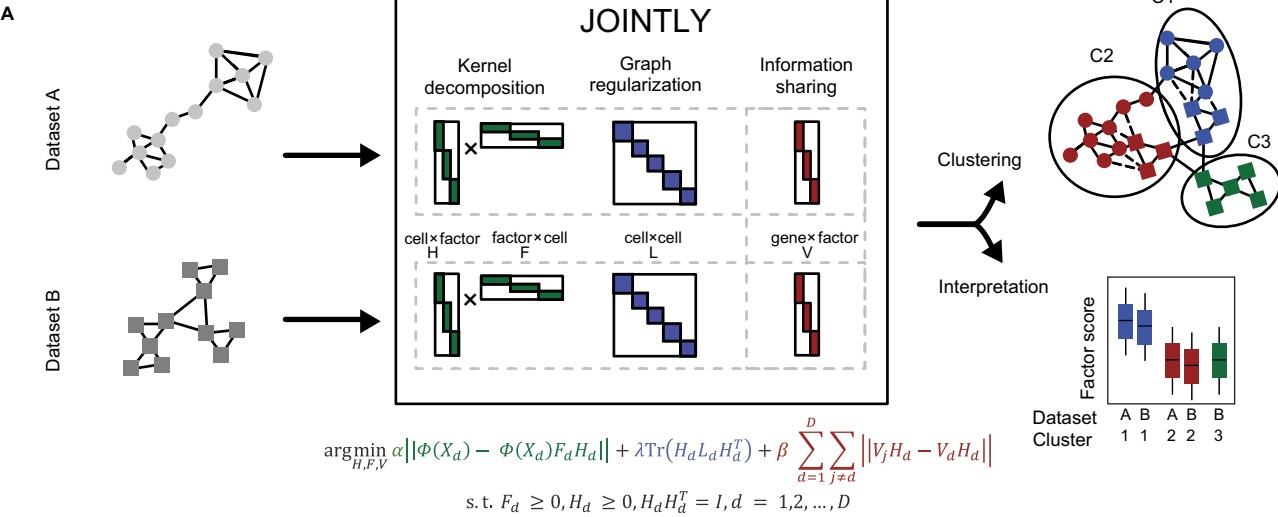

$$\underset{H,F,V}{\arg\min} \, \alpha \left\| \Phi(X_d) - \Phi(X_d)F_d H_d \right\| + \lambda \text{Tr}\left(H_d L_d H_d^T\right) + \beta \sum_{d=1}^{D} \sum_{j \neq d} \left\| V_j H_d - V_d H_d \right\|$$

$$\text{s. t. } F_d \geq 0, H_d \geq 0, H_d H_d^T = I, d = 1, 2, \ldots, D$$

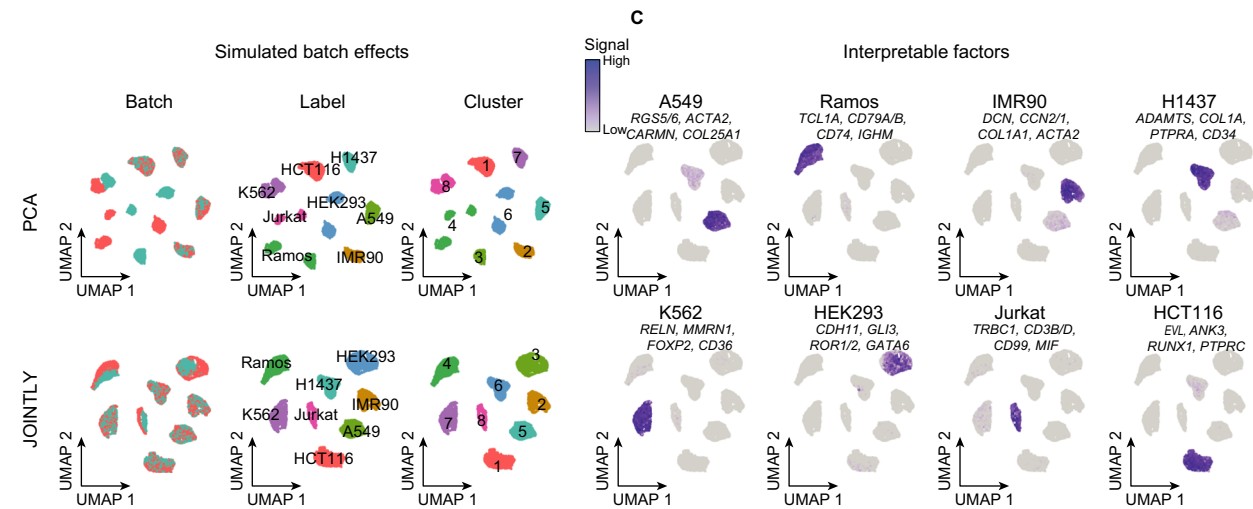

**Fig. 1 | JOINTLY clusters single-cell RNA-seq and single-nucleus RNA-seq (sxRNA-seq) datasets without explicit integration. A** Schematic illustration of JOINTLY. JOINTLY uses a hybrid linear and non-linear non-negative matrix factorisation to optimise reduced dimensional spaces, which allows for clustering across datasets and interpretation to discover conserved and active biological processes. The gene expression matrix ($X$) for each dataset ($d$ in $D$) is decomposed into lower rank matrices ($H$ and $F$) in a mapped higher or infinitely dimensional space ($\Phi$). Graph regularisation is applied using the graph Laplacian $L$. Interpretable factors $V$ constrain JOINTLY to patterns that are generalisable across datasets. $\alpha$, $\beta$ and $\lambda$ weigh components of the loss function. Subject to (s.t.) non-negativity constraints on the $F$ and $H$ matrix. Tr is the trace matrix. $I$ is the identity matrix. **B, C** UMAP for 8 cell lines[11] with simulated batch effects (see "Methods") based on reduced dimensional spaces calculated using PCA or JOINTLY. The UMAPs are coloured by batch, cell line, and cluster (**B**) or by selected gene modules derived from the interpretation JOINTLY (**C**). Source data are provided as a Source Data file.

This kernel is factorised using kernel non-negative matrix factorisation[10] into two low-rank matrices (factor by cell and cell by factor) regularised by shared nearest neighbours graphs. The third and final low-rank matrix is a linear feature matrix (factor by gene), which is minimised to generalise across all batches. This ensures that learned factors, which describe the gene expression space in each dataset, are characterised by the same genes across batches facilitating both joint clustering and interpretation.

To evaluate the ability of JOINTLY to cluster scRNA-seq datasets with batch effects, we initially analysed a simple dataset consisting of eight purified cell lines[11]. We randomly split the dataset into two batches and simulated non-linear and cell type-dependent batch effects (see "Methods"). The clusters identified using JOINTLY recover the original cell types, whereas clustering on the unintegrated dataset wrongly assigned one batch of Ramos cells to the Jurkat cluster (Fig. 1B). In addition, JOINTLY also identifies interpretable factors that represent gene modules driving clustering. These modules are highly cell line-specific containing genes relevant to cellular function (Fig. 1C, Supplementary Fig. S1A). For example, Decorin (*DCN*), a fibroblast marker gene, is contributing to the module highly specific for fibroblast-derived IMR90 cells, while Cluster of Differentiation 3B (*CD3B*), a T-cell coreceptor, is contributing to the module highly specific for T-cell derived Jurkat cells. The gene modules are generally not strongly correlated to each other within a single batch, but highly correlated across batches (Supplementary Fig. S1B) suggesting that the modules represent different but reproducible gene programs. To assess how these modules are affected by the presence of batch effects, we analysed the same datasets without the simulated batch effects and compared the detected gene modules. We found that the modules in the datasets with and without simulated batch effects are highly conserved (Supplementary Fig. S1C) indicating that JOINTLY discovers reproducible gene programs independent of batch effects.

## JOINTLY achieves state-of-the-art clustering performance

To compare the joint clustering performance of JOINTLY to existing methods, we applied JOINTLY and eight state-of-the-art integration methods[1–3,7,12–14] to a sxRNA-seq dataset from the human lung containing ~10,000 cells distributed over six batches[4]. We reannotated the dataset automatically using a similar public dataset[15] and support vector classification (see "Methods"), which in recent tests has been shown to have very good performance for label transfer[16,17]. We chose to reannotate the data, rather than using labels defined by the original authors, to avoid any potential biases in favour of the clustering and batch integration methods used in the original publication. In this dataset, several methods, including JOINTLY, reach an adjusted rand index (ARI) in the range of 0.89−0.91 (Fig. 2A). Notably, among the tested interpretable methods, only JOINTLY reaches this state-of-the-art performance level. No method achieves perfect performance since they all fail to separate endothelial subtypes (blood vessel, vein, and lung microvascular), as well as myeloid cell types (classical monocytes, macrophages, and dendritic cells), but do correctly group them.

Next, we expanded the benchmark by including an additional 4 integration tasks containing between 8444 and 46,993 cells distributed over five to 18 batches[4,18–21] with labels transferred from similar datasets[4,20–24]. Summarised across all tasks, JOINTLY ranks second only superseded by scVI, while the other interpretable methods rank eighth and ninth, respectively (Fig. 2B, Supplementary Fig. S2A, see Source Data). In joint clustering, all datasets may not be equally well clustered, or some datasets may dominate the integration due to for example a difference in the number of cells. To assess if there is a performance imbalance between datasets, we evaluated the clustering performance in each dataset after joint clustering. Together with scVI and Scanorama, JOINTLY ranks first in the worst dataset of each task (Fig. 2B) showing that JOINLY has a highly balanced performance across datasets. Taken together, this demonstrates that JONTLY

effectively captures latent gene expression patterns, which are related to cellular identity, and generalises well across all datasets.

In addition to clustering performance, we also evaluated batch and cell type mixing using local inverse Simpson's index[2] (LISI) and the average silhouette width (ASW) (Fig. 2C). The cell type LISI (cLISI) assesses cell type mixing; the number of closest neighbours for each cell that are of a different cell type. All methods perform approximately equally well, including naïve unintegrated analysis. The cell type ASW (cASW) assesses how separated cell types are; the average distance to another cell of the same cell type compared to the average distance to another cell of a different type. JOINTLY ranks fifth followed by scVI and Scanorama. The integration LISI (iLISI) assesses batch mixing; the number of closest neighbours for each cell that is of a different batch. JOINTLY ranks seventh after scVI. Finally, batch ASW (bASW) assesses how separated batches are; the average distance to another cell of the same cell type in the same batch compared to the average distance to another cell of the same cell type in a different batch. For this metric, JOINTLY ranks first followed by LIGER and then scVI, scGPT, and Scanorama. In summary, JOINTLY achieves state-of-the-art joint clustering performance and has similar trade-offs as scVI and Scanorama in terms of cell type and batch separation and mixing.

One of the differences between JOINTLY and other methods is that JOINTLY, by default, uses consensus PCA (see "Methods") as a basis for integrating samples. Therefore, the initial decomposition of each sample depends on the other samples. Thus, integration performance is dependent on the number and similarity of input samples. To evaluate the extent of this dependency, we removed all sample combinations from the human liver dataset resulting in 25 subsamples with two to four batches. We found no difference in the ARI, in the average cell type and integration LISI nor the cell type and batch ASW for subsamples with two to four batches compared to the full dataset with five batches (Fig. 2D−H, Supplementary Fig. 2B), although we did find that with fewer batches, the standard error of the mean increases, indicating that integration performance becomes more variable. We also evaluated how well the nearest neighbours are conserved between each subsample and the full dataset and found that approximately 50−60% of the nearest neighbours are conserved with a minor decrease with fewer batches, which is similar in range to the variation between different integrations on the full dataset (Fig. 2I). Collectively, this indicates that the performance of JOINTLY is not strongly dependent on the number of input datasets.

## JOINTLY retains biological variation across conditions

An important use case for joint clustering is multi-sample, multi-condition datasets, such as studies comparing several healthy and diseased individuals. In such a dataset, the aim is to make inferences on the level of conditions, while controlling for sample-to-sample variation. It is especially challenging to perform joint clustering in this type of dataset since both batch effects and biological variability between conditions contribute to intra-sample variability. Thus, we cannot assume a priori that all samples should overlap. To evaluate the ability of JOINTLY and alternative methods to retain biological variability between conditions, while removing batch effects between samples, we created 10 multi-sample, multi-condition datasets by pairwise combination of the 5 datasets used for benchmarking. For each integration method, we evaluated the integration and cell type LISI for each tissue, as well as the integration LISI across tissues. We found JOINTLY consistently achieves a high overall ranking (Fig. 3A, see Source Data). Generally, JOINTLY balances the preservation of biological variation, while removing batch effects within each tissue, with obtaining a good separation across tissues. As an example, we analysed the mixture of Pancreas and Lung in more detail and found upon visualising the data, that all methods, except LIGER and Seurat, successfully keep the majority of the two datasets unmixed. Of the remaining methods, JOINTLY, Scanorama, Harmony, and scVI obtain

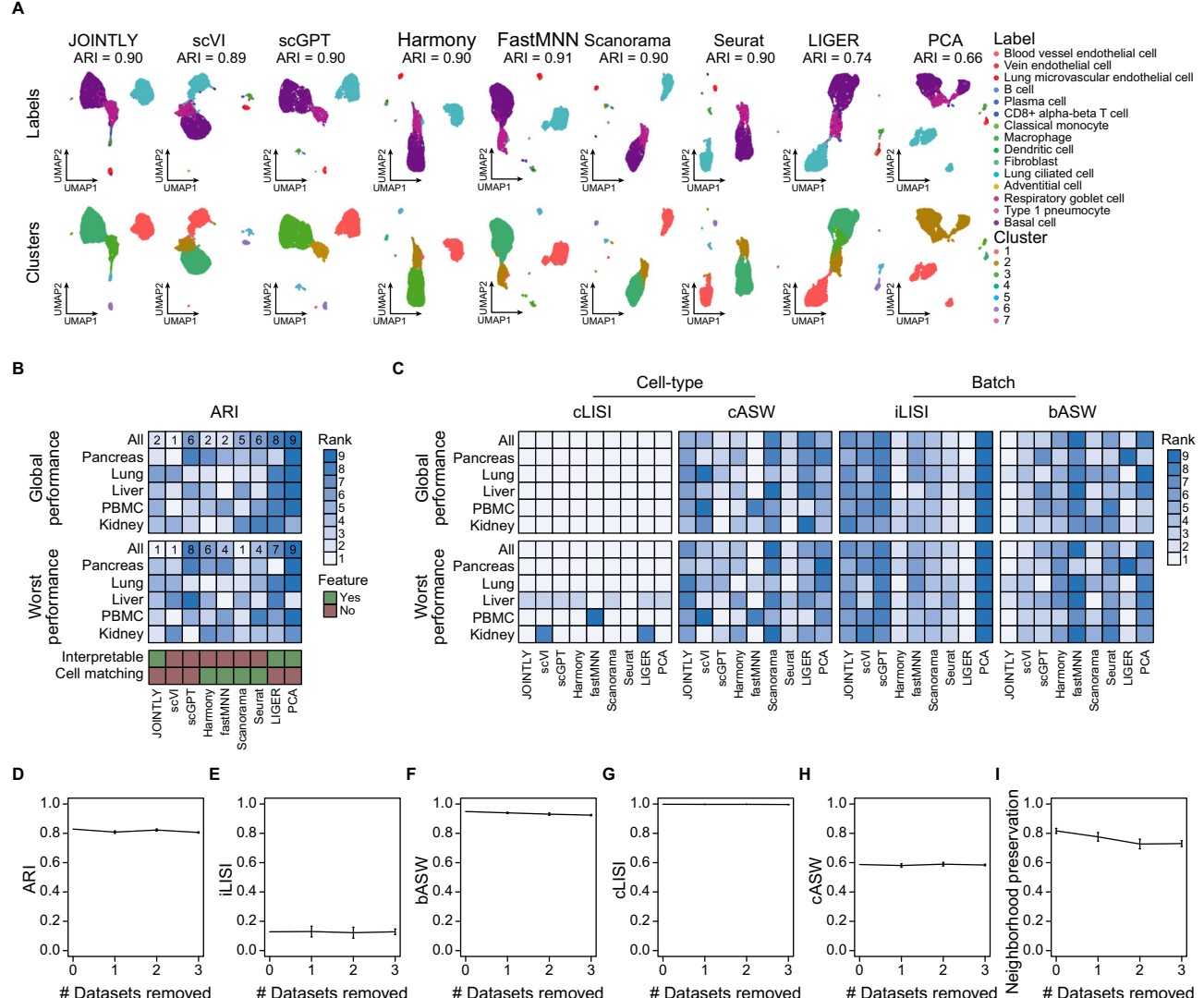

**Fig. 2 | JOINTLY performs on par with competing batch integration and clustering methods. A** UMAP based on embedded spaces calculated by the indicated methods and coloured by transferred cell type labels and clusters in the Lung dataset, as indicated in the figure. Heatmap showing ranks aggregated across all datasets (global) and for the worst dataset (worst) based on the adjusted rand index (ARI) between clusters and cell types (**B**) as well as ranks based on batch and cell type separation and mixing evaluated using cell type local inverse Simpson's index (cLISI), cell type average silhouette width (cASW), integration LISI (iLISI) and batch ASW (bASW) (**C**) for the indicated methods and datasets. Line charts showing ARI (**D**), iLiSI (**E**), bASW (**F**), cLISI (**G**), and cASW (**H**) for the full dataset (zero datasets removed), and the average across all possible subsamples removing between one and three datasets from the Lung dataset where $n_0 = 5$, $n_1 = 5$, $n_2 = 10$ and $n_3 = 10$ unique sample combinations. Error bars show the standard error of the mean. **I** Line chart showing the average fraction of conserved neighbours. For the full dataset (zero datasets removed), the average was calculated from the fraction of conserved neighbours between different runs of JOINTLY on the full dataset and a reference run on the full dataset where $n_0 = 5$. For subsamples (between one and three datasets removed), the average was calculated from the fraction of conserved neighbours between the integration of the subsampled dataset and the reference run on the full dataset, where $n_1 = 5$, $n_2 = 10$ and $n_3 = 10$ unique sample combinations. Error bars show the standard error of the mean. Source data are provided as a Source Data file.

good integration of the samples, while separating within-tissue cell types (Fig. 3B and Supplementary Fig. S3A, B). Among these four methods, we found subtle differences. For example, Harmony mixes the endothelial cells across tissues, while JOINTLY, scVI, and Scanorama place them close to each other in embedded space but retain tissue-specific clusters. This prompted us to investigate the similarity of endothelial cells between tissues. Initially, we identified shared marker genes and found that the shared marker genes contain several known endothelial markers, such as lymphatic vessel endothelial hyaluronan receptor 1 (*LYVE1*) and melanoma cell adhesion molecule (*MCAM*) (Fig. 3C) highlighting that both populations are bona fide endothelial cells. However, differential expression analysis between the two populations revealed a large set of 3529 differentially

expressed genes that are similarly expressed across samples within each tissue (Fig. 3D). We submitted lung- and pancreas-specific endothelial markers to enrichment analysis using a database of transcription factor targets and found an almost complete dichotomy (Fig. 3E) suggesting that distinct transcriptional programs and regulatory mechanisms are shaping endothelial cells from the two tissues. As an example of different biological processes shaping the cells, we found that interferon signalling is specifically high in the lung-derived endothelial cells (Fig. 3F). Taken together, this strongly suggests that lung-derived and pancreas-derived endothelial cells represent different states of the same cell type. JOINTLY correctly retains this variability indicating that JOINTLY is robust against deletion of condition-specific variation in multi-sample, multi-condition datasets.

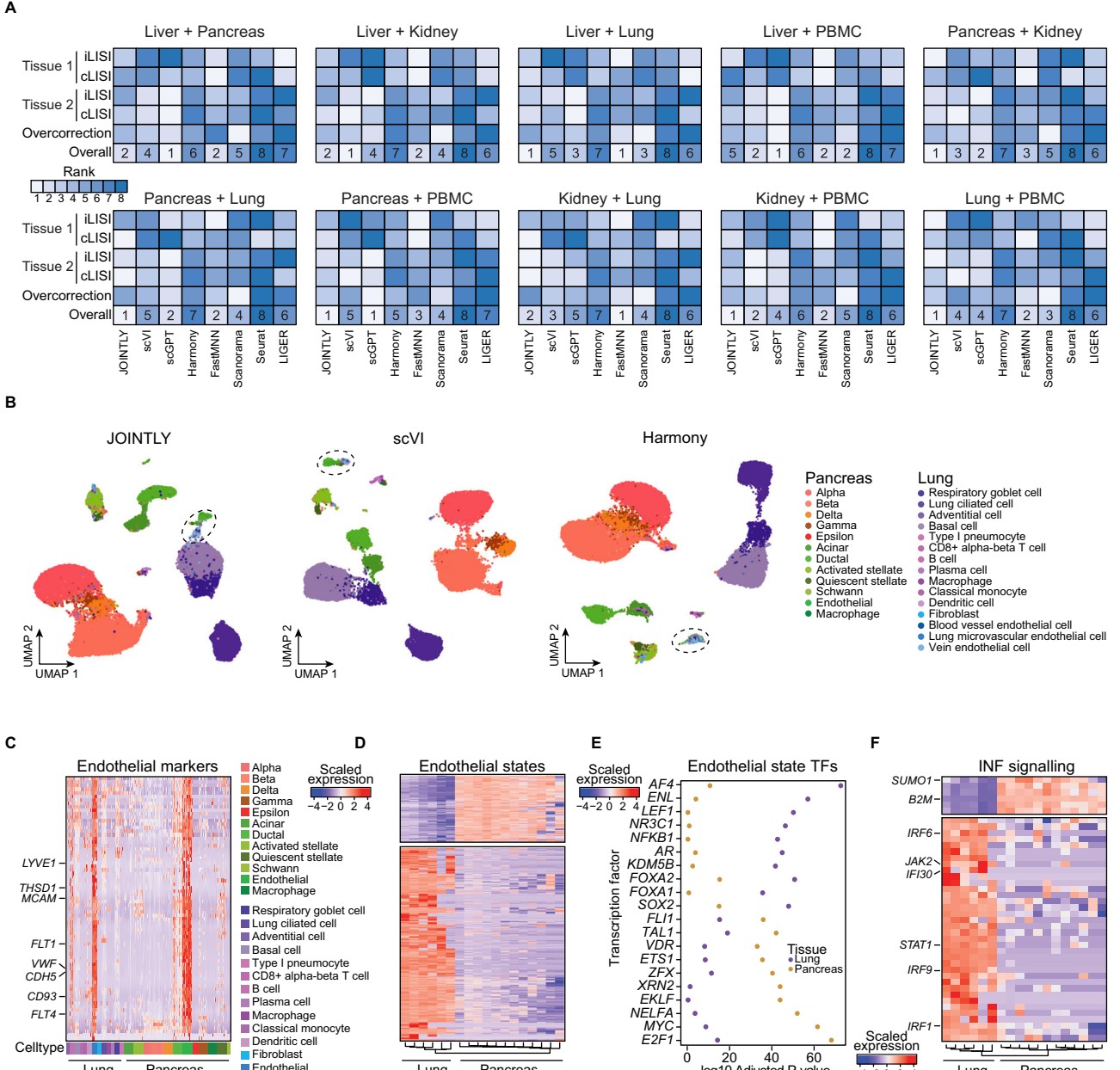

**Fig. 3 | JOINTLY is robust to cell state deletion. A** Heatmap showing ranked integration metrics (iLISI and cLISI within each tissue and over-correction, which is iLISI between tissues) for mixtures of the indicated tissues. **B** UMAP based on embedded spaces calculated by the indicated methods and coloured by trans-ferred cell type label (see "Methods") for the combined Lung and Pancreas dataset. **C** Heatmap showing normalised and scaled per-donor pseudo-bulk expression of marker genes shared between pancreas-derived and lung-derived endothelial cells. **D** Heatmap showing normalised and scaled per-donor pseudo-bulk expression of differentially expressed genes between pancreas-derived and lung-derived endo-thelial cells. **E** Scatterplot showing the -log₁₀ Benjamini-Hochberg-corrected *p*-values of Enrichr fisher's exact test of the enrichment of transcription factor target gene among differentially expressed genes between pancreas-derived and lung-derived endothelial cells. **F** Heatmap showing normalised and scaled per-donor pseudo-bulk expression of interferon signalling genes in pancreas-derived and lung-derived endothelial cells. Source data are provided as a Source Data file.

## Interpretable factors augment cell type annotation and uncover active biological processes

Both JOINTLY and scVI achieve state-of-the-art performance in joint clustering and are both robust against over-correction. However, one defining difference between the two approaches is that JOINTLY is intrinsically interpretable. To evaluate the value of JOINTLY's inter-pretable factors, we reanalysed a dataset from human white adipose tissue subset to whole tissue single-nucleus RNA-seq on visceral adi-pose tissue from nine donors[25] using JOINTLY to integrate across donors and different visceral adipose tissue sources (Supplementary Fig. 4A, B). We identified 13 unique cell types (Fig. 4A), each of which is

enriched for known marker genes (Fig. 4B) and overlaps the cell types identified by the original authors (Supplementary Fig. 4C). We scored the gene modules identified by JOINTLY in each cell and found that the modules are enriched in one cell type or a group of related types (Fig. 4C, see Source Data). Inspecting the modules, we found that they contain many genes associated with cellular identity. For example, factor 13, which is specific to adipocytes, contains genes such as cluster of differentiation 36 (*CD36*), diacylglycerol O-acyltransferase 2 (*DGAT2*), lipoprotein lipase (*LPL*), perilipin 5 (*PLIN5*), adiponectin (*ADIPOQ*), acyl-CoA synthetase long chain 1 (*ACSL1*), and peroxisome proliferator-activated receptor γ (*PPARG*), and factor 7, which is

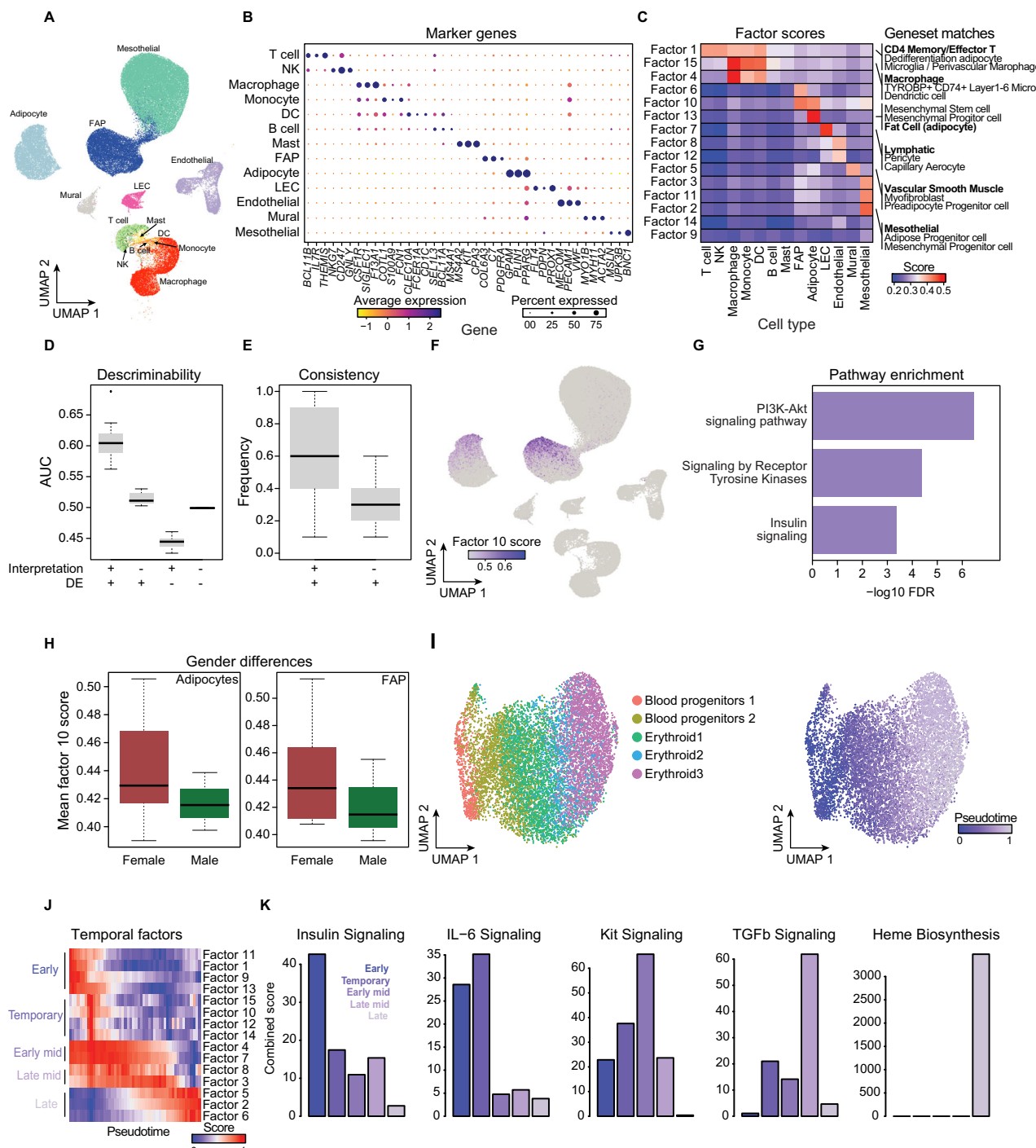

specific to lymphatic endothelial cells (LEC), contain genes such as *LYVE1*, reelin (*RELN*), multimerin 1 (*MMRN1*), and neuropilin 2 (*NRP2*). To test if the modules and associated genes can be used for cell type annotation, we performed gene set enrichment analysis using the modules and databases of cell type-specific marker genes and found that for several cell types, the correct label was identified in the top three best hits per factor (Fig. 4C, right annotation, see Source Data).

We compared the modules identified by JOINTLY to the list of marker genes for each cell type and found a high overlap between modules and marker genes (Supplementary Fig. 4D). However, we found that the marker genes, which are also module genes, have a significantly higher area under the curve (AUC) than marker genes, which are not in the most enriched module (Fig. 4D), indicating that module genes are more discriminatory between cell types than marker

genes. Similarly, we found that module genes, which are not marker genes, have a significantly lower AUC than genes, which are neither marker nor module genes (Fig. 4D) indicating these module genes are markers of other cell types. Finally, we evaluated the consistency of markers across batches by finding marker genes in each batch independently and found that marker genes, which are also module genes, are more often markers in multiple batches compared to marker genes, which are not module genes (Fig. 4E). Collectively, this indicates that module genes are highly discriminatory between cell types, and more so than marker genes.

In addition to cell type-specific modules, we also observed shared gene modules with non-uniform distribution within enriched cell types. For example, factor 10, which scores high in a subset of adipocytes and fibro-adipogenic progenitors (FAPs) (Fig. 4F). We

**Fig. 4 | Interpretable factors uncover active biological processes. A** UMAP based on JOINTLY embeddings of 73,118 adipose tissue cells from 9 batches coloured by cell type annotations[25]. Cells were annotated based on marker gene expression. **B** Dot plot showing expression levels and frequency for selected marker genes for each identified cell type. **C** Heatmap showing average module scores using gene modules derived from JOINTLY factors across the identified cell types. Annotations on the right show the top 3 most enriched cell types identified by pathway analysis. Correct matches between enrichment results and manual curation are highlighted in bold. **D** Boxplots showing area under the curve (AUC) for genes identified through interpretation of JOINTLY or through differential gene expression analysis between cell types. The centre represents the median AUC across the 13 cell types and whiskers indicate 1.5 times the interquartile range above or below the 75% and 25% quantiles, respectively. **E** Boxplot showing the fraction of marker genes that are identified as marker genes in at least two batches stratified by whether the marker gene is also identified through interpretation of JOINTLY. The centre represents the median fraction across the 13 cell types and whiskers indicate 1.5 times the

interquartile range above or below the 75% and 25% quantiles, respectively. **F** UMAP based on JOINTLY embeddings of 73,118 adipose tissue cells from 9 batches coloured by module score of genes assigned to JOINTLY factor 10. **G** Barplot showing the $-\log_{10}$ FDR-corrected $p$-values from Enrichr fisher's exact test of selected pathways that are enriched for genes assigned to JOINTLY factor 10. **H** Boxplot showing module scores for genes assigned to JOINTLY factor 10 across adipocytes and FAP stratified by donor sex. The centre represents the mean and whiskers indicate 1.5 times the interquartile range above or below the 75% and 25% quantiles, respectively. **I** UMAP based on JOINTLY embeddings of cells undergoing erythropoiesis during mouse gastrulation[28] from 3 batches coloured by cell type annotations. **J** Heatmap showing the average module score for all JOINTLY factors across cell bins. Cells were binned in 50 groups based on pseudo-time (calculated using scVelo[27]). Modules were grouped based on their temporal profile. **K** Barplots showing the combined score of selected pathways identified by pathway enrichment analysis of groups of modules (see **J**). Source data are provided as a Source Data file.

hypothesised that such factors may represent distinct transcriptional states or programs. Pathway analysis revealed that factor 10 contains genes involved in insulin signalling and sensitivity (Fig. 4G) suggesting that this factor marks cells with high insulin signalling capacity. Stratifying adipocytes and FAPs by sex revealed that factor 10 scores high specifically in females indicating that females may have higher adipose tissue insulin sensitivity than men consistent with existing literature[26] (Fig. 4H).

In addition to discrete cell types, sxRNA-seq can also be used to probe continuous biological processes, such as differentiation. To test JOINTLY in this setting, we applied JOINTLY and scVelo[27] to three batches of cells undergoing erythropoiesis from an atlas of mouse gastrulation[28]. We found that JOINTLY orders cells along cell types and along latent time in a comparable manner to scVelo (Fig. 4I, Supplementary Fig. 4E). Interpretation of JOINTLY revealed that the JOINTLY factors have different temporal profiles (Fig. 4J). Pathway analysis of genes associated with early, temporary, early-mid, late-mid, or late latent time revealed signalling cascades with different timing, consistent with existing literature on erythropoiesis (Fig. 4K). Collectively, this suggests that the interpretable factors learned by JOINTLY can be used to guide the analyst toward annotating their datasets, recovering temporal dynamics, and discovering new biological insights.

## Building a white adipose tissue reference atlas using JOINTLY

Finally, having established that JOINTLY achieves high clustering performance, we set out to generate a reference atlas for white adipose tissue as a vignette of a use-case of JOINTLY to generate a community resource. Integration of tissue atlases has different requirements than integration of multiple samples in a single dataset. The datasets used to construct tissue atlases often are more heterogeneous and have stronger batch effects, as the datasets are often generated using a multitude of technologies across several different laboratories. Furthermore, a tissue atlas is often aimed towards being a resource for a community, and it is therefore beneficial to create tissue atlases using methods with transfer learning capabilities. This enables the atlas to grow as new data is added, and it allows researchers to use the atlas to contextualise their data without sharing raw data. Recent benchmarking efforts have shown that semi-supervised integration using scANVI is among the best-performing methods in this space[4]. However, to apply semi-supervised integration and build a state-of-the-art tissue atlas, cell type labels are required.

We uniformly annotated cells from white adipose tissue from six independent sxRNA-seq studies[15,25,29–32] by applying JOINTLY to each study separately (Fig. 5A). Subsequently, we created an expandable tissue atlas, termed WATLAS, by integration using scANVI[33], totalling ~300,000 cells from both visceral (VAT) and subcutaneous adipose tissue (SAT) depots. In the WATLAS, we were able to recover 17 major cell types in WAT, and 43 total subtypes (Fig. 5B, Supplementary

Fig. 5A, B). Our labels for cell types and subtypes are highly consistent with the original author labels (Supplementary Fig. 5C) and all cell types and subtypes are supported by robust gene signatures (Supplementary Fig. S5D, see Source Data). We have made the atlas, annotations, and metadata openly and freely available[34] as well as the model weights for transfer learning[35]. To highlight the richness of this resource for hypothesis generation, we investigated how obesity affects the composition of the white adipose tissue. Controlling for sex and depot, we found that obesity is associated with a trend towards a decrease in the fractional number of smooth muscle cells and fibro-adipogenic progenitors (FAPs), driven by $CXCL14^+$ and $PPARG^+$ FAPs, and a trend towards an increase in $LPL^+$ and $LYVE1^+$ macrophages as well as all subtypes of adipocytes, except PRSS23$^+$ adipocytes (Fig. 5C). An increase in adipocytes and a decrease in FAPs could suggest an increased rate of adipogenesis in obesity in humans, similar to what we have previously reported in obese mice[36].

The number of transcriptional states of mature adipocytes is currently being debated in the field with studies reporting between three and seven subtypes[25,32,37]. Adding to this debate, a recent integrated analysis of sxRNA-seq studies found low batch integration for adipocytes and inconsistent enrichment of marker genes across studies[38] suggesting either low adipocyte heterogeneity or inadequate power to detect it due to either technical or analytical challenges. In the WATLAS, we annotated four different adipocyte populations; $DCN^+$ adipocytes are the most distinctive population marked by genes normally associated with expressed in fibro-adipogenic progenitors, such as $DCN$, apolipoprotein D ($APOD$) and Lumican ($LUM$) suggesting they may represent newly differentiated adipocytes. The second most distinct population is the $PRSS23^+$ adipocytes, which is similar to the hAd3 population defined by the Rosen group[25] and express several sulfo-transferases, such as sulfotransferase family 2B member 1 ($SULT2B1$) and glutamate NMDA receptor components, such as glutamate ionotropic receptor NMDA type subunit 2A ($GRIN2A$) (see Source Data). $CLSNT2^+$ adipocytes resemble the hAd5 population defined by the Rosen group[25] and express genes associated with insulin signalling and sensitivity, such as ectonucleotide pyrophosphatase phosphodiesterase 1 ($ENPP1$). Finally, $DGAT2^+$ adipocytes, which resemble hAd4 as defined by the Rosen group[25] express lipolysis and lipogenesis genes, such as monoacylglycerol o-acyltransferase 1 ($MOGAT1$) and diacylglycerol o-acyltransferase 2 ($DGAT2$). All four populations have variable, but high expression of established adipocyte markers compared to non-adipocyte cells (Fig. 5D) suggesting that they all represent bona fide adipocytes. We investigated the similarity of gene expression programs across populations, depots, and studies by identifying marker genes for each population and scoring the enrichment of all marker gene modules in each cell. We found that modules are enriched in the population of origin across studied and depots, except for SAT-derived $PRSS23^+$ adipocytes from the Lumeng group,

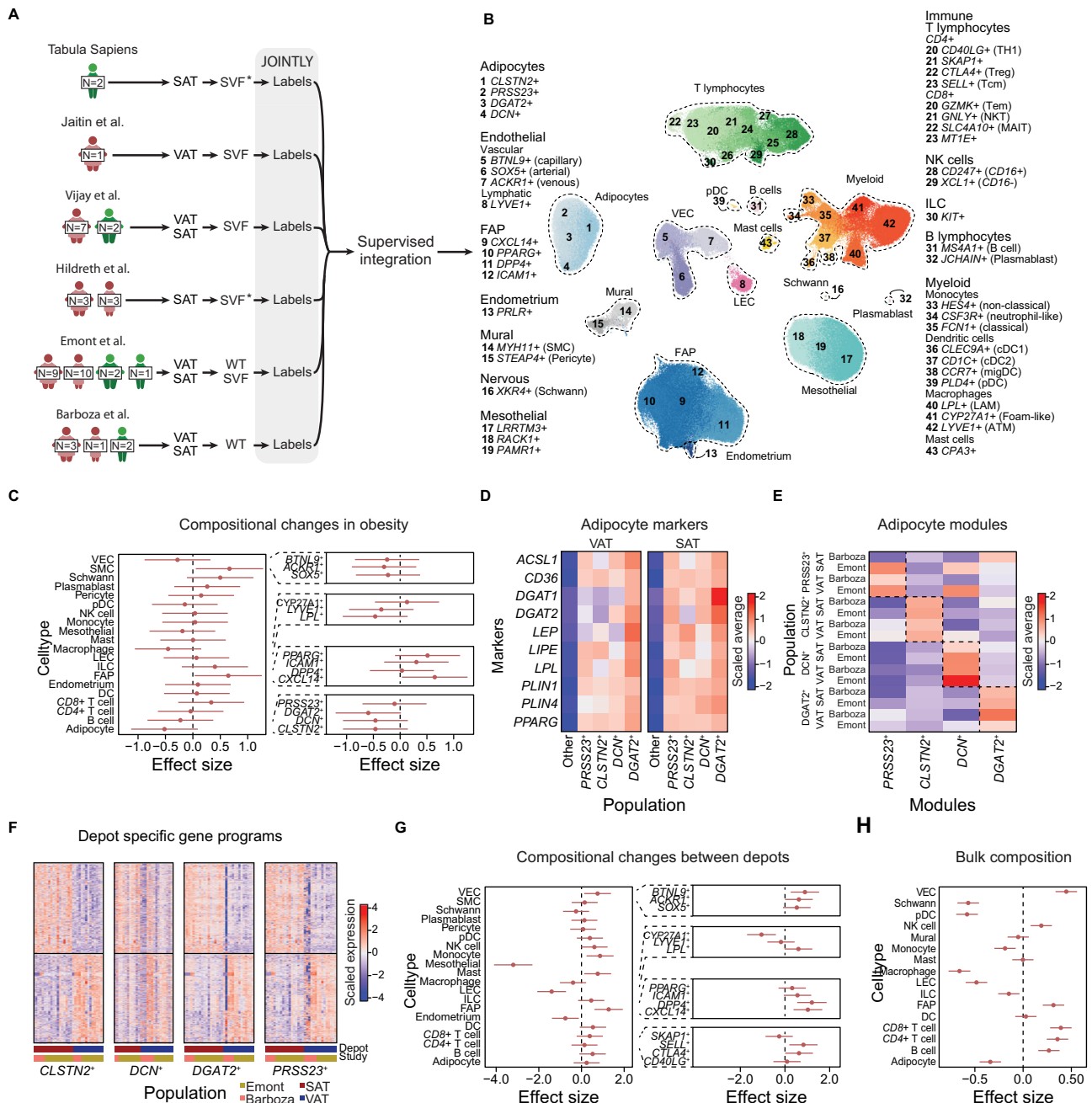

**Fig. 5 | Building a white adipose tissue atlas with JOINTLY. A** Illustration of the datasets and workflow used to generate the white adipose tissue atlas (WATLAS). The colour of the characters indicates the donor sex (green = male, red = female), visceral adipose tissue (VAT) and subcutaneous adipose tissue (SAT) indicates the depot origin, stromal vascular fraction (SVF) and whole tissue (WT) indicate the cell fractions used. * Indicate a fraction was enriched for. **B** UMAP based on scANVI[33] embeddings coloured by cell type labels. Cells were annotated based on marker genes and clustering of each individual dataset with JOINTLY. Subsequently, scANVI was trained using the JOINTLY labels. **C** Scatterplot showing compositional changes in cell type abundances between individuals with or without obesity, corrected for depot and sex in 46 donors. Broad cell types in the left panel, and states of high-lighted cell types in the right panel. Centres represent the mean effect size and error bars represent the 95% confidence intervals. **D** Heatmap showing normalised and standardised average expression of adipocyte markers in adipocyte sub-populations and all other cell types stratified by the depot. **E** Heatmap showing

normalised and standardised average module scores in adipocyte subpopulations stratified by depot and study. Modules were generated using differentially expressed genes in the indicated adipocyte subpopulations. **F** Heatmap showing normalised and scaled per-donor pseudo-bulk expression levels of differentially expressed genes between depots stratified by adipocyte subtypes, study, and depot. **G** Scatterplot showing compositional changes in cell type abundances between adipose tissue depots, corrected for sex and weight status in 46 donors. Broad cell types in the left panel, and states of highlighted cell types in the right panel. Centres represent the mean effect size and error bars represent the 95% confidence intervals. **H** Scatterplot showing compositional changes in cell type abundances between adipose depots, corrected for sex using deconvoluted cell type abundances from 1293 donors. Centres represent the mean effect size and error bars represent the 95% confidence intervals. Source data are provided as a Source Data file.

which has stronger enrichment for a *DGAT2*[+] signature (Fig. 5E). The expression patterns of individual marker genes, although less clear, support that the four populations reported here replicate between studies and across depots (Supplementary Fig. 6A).

However, there are depot-specific differences in the expression level of established marker genes as well as in the enrichment of gene modules suggesting that the depot shapes the gene expression profile of the adipocytes. A deeper analysis of depot-specific gene programs in adipocyte subpopulations revealed that depot differences are shared between subpopulations and studies (Fig. 5F), although expression is sparser in *DCN*[+] adipocytes compared to other adipocyte subpopulations, which we attribute to the low number of *DCN*[+] adipocytes. Taken together, this indicates that the niche effects from the depot affect all adipocyte populations similarly. However, we found that different depots are characterised by different white adipose tissue composition. VAT is enriched for mesothelial cells, macrophages overall, and especially *LPL*[+] macrophages (that are similar to *TREM2*[+] macrophages), lymphatic endothelial cells (LECs), and endometrium, whereas SAT is enriched for vascular endothelial cells (VECs), *DPP4*[+] and *CXCL14*[+] FAPs and several immune cell types (Fig. 5G, see Source Data). To support these claims, we deconvoluted 562 VAT and 731 SAT bulk RNA-seq samples from GTEx at medium label resolution using WATLAS and evaluated depot differences in adipose tissue composition, excluding mesothelial cells, which are only present in VAT. The average deconvoluted cell type fractions in both VAT and SAT was highly correlated with the observed cell type fractions observed in the WATLAS (Supplementary Fig. 6B) and several trends from the single-cell dataset are conserved including enrichment of VEC, FAPs, and T cells in SAT and enrichment of macrophages and LECs in VAT (Fig. 5H). These observations are consistent with reports that find VAT has a higher content of macrophages in mice[39] and humans[40,41], while SAT has been reported to have a higher content of adipocyte progenitors in humans[40] and a higher proportion of *CD4*[+]T cells[42]. In the literature, there are conflicting reports concerning vascularisation, capillary density, and endothelial cells. Some studies find an increase in VAT compared to SAT[43–45], while other studies find no difference 45 or the opposite[46]. This suggests that the difference in vascularity between depots depends on the biological context.

## Discussion

To deeply characterise the transcriptional state of cells in the human body using single-cell or single-nucleus RNA-sequencing (sxRNA-seq), and evaluate how diverse biological processes, such as development and disease, alter those states, it is necessary to distinguish between biological and technical variation. Here, we introduce JOINTLY, a hybrid linear and non-linear matrix factorisation-based method for joint clustering of sxRNA-seq. JOINTLY aims to capture shared features across datasets without explicitly harmonising them. To that end, JOINTLY defines a reduced dimensional space, using consensus PCA, which is based on shared axes of variation supplemented with the batch-specific axis of variation, that describes most of the variation across all datasets. This choice is appropriate for datasets, where the major axes of variation are assumed to be similar, such as biological replicates. For more complex datasets, where all samples do not necessarily share the same axes of variation, such as multi-condition, multi-sample datasets, we advise users to use regular PCA in JOINTLY to ensure that the most important axes of variation are captured. Cellular similarity is estimated in this reduced space using a data-adaptive heat-based kernel[9] as well as a shared nearest neighbourhood graph. These measures are used to learn a reduced dimensional space, which reconstructs non-linear cell-to-cell similarity in each dataset, as well as the linear gene expression space across datasets. This allows for joint clustering of the data without explicitly assuming any overlap between datasets, and it allows for the interpretation of the model to discover genes' contribution to clustering. We evaluate JOINTLY and

compare its performance to eight state-of-the-art batch integration methods[1–3,7,12–14] on 52 sxRNA-seq samples from five tissues. The chosen methods are widely applied in the field, and are, like JOINTLY, unsupervised integration methods. We chose to only include unsupervised methods, as systematic benchmarking has shown that (semi)supervised integration methods often outperform their unsupervised counterparts[4]. JOINTLY performs on par with state-of-the-art batch integration tools, such as scVI[3] and Harmony[2], in clustering tasks and has a similar trade-off between biological heterogeneity and batch mixing as scVI. In line with a recent benchmark[47], we found that JOINTLY and several task-specific models, outperformed scGPT, a foundational single-cell RNA-sequencing model. As a future perspective, we envision that the performance of JOINTLY can be even further improved by initialising the algorithm using cell type labels.

A critical and challenging task for joint clustering is multi-condition, multi-sample datasets, where both batch effects, which should be removed, and biological variation introduced by the condition, which should be retained, contribute to intra-sample variance. To assess how the different methods perform in this setting, we generated an artificial multi-condition, multi-sample dataset by mixing single-cell RNA-sequencing samples from the lung and the pancreas. We found that JOINTLY, scVI, and Scanorama retain biological variation across conditions, while removing batch effects, suggesting that these methods are preferable for analysing multi-condition, multi-sample datasets, such as cohort studies of healthy and diseased individuals.

In our benchmark, we noted that although JOINTLY, scVI[3], and Harmony[2] generally have the best performance, there is substantial task-to-task variation. Across tasks, JOINTLY achieves ranks between second and fifth, while both scVI and Harmony rank between first and seventh. This indicates that no method is always the best across all tasks, and therefore, that it is critical to evaluate the performance of a chosen method on any new integration task. This can be accomplished using various integration metrics, such as those formalised in scIB[4]. However, such metrics do not provide insight into the genes or features driving integration and clustering. By design, JOINTLY is interpretable meaning that it learns which gene modules contribute to integration and clustering. Analysis of these gene modules gives the analyst an immediate insight into the dataset, allowing the analyst to compare genes driving clustering to known marker genes, thereby evaluating the biology of integration. Based on a cell line dataset with simulated batch effects, we found that the interpretable factors generated by JOINTLY are highly insensitive to batch effects. To further evaluate the interpretable factors, we processed a dataset from human visceral white adipose tissue consisting of nine samples. We found that the interpretable factors can be used to predict cell type labels and to re-discover a known sexual dimorphism in insulin signalling in adipocytes and fibro-adipogenic progenitors. Compared to regular cluster-based differential expression analysis, the genes identified through interpretation of JOINTLY are more discriminative, and more conserved between batches. An additional advantage of the interpretable factors is that they not limited to being enriched in a cluster or cell type but can be shared between several cell types that share transcriptional programs, or specific to certain conditions. This allows the analyst to discover active biological programs that would have been missed by cluster-based DE analysis. Taken together, this indicates that in addition to helping the analyst evaluate clustering, the interpretable factors can also be used to guide the functional annotation of a dataset and to discover important active biological processes.

Finally, we use JOINTLY to create a white adipose tissue reference atlas (WATLAS) by processing and labelling six independent datasets with JOINTLY and subsequently, integrating across datasets using scANVI, a semi-supervised integration method. This yields an expandable atlas available as a resource for the community to explore[34] and use for contextualisation of new datasets with transfer learning[35]. In the WATLAS, we identify 46 distinct cell states, with

unique and specific gene signatures. Our WATLAS covers a high dynamic range of cell abundances, from highly abundant adipocytes to ultra-rare populations, such as Schwann cells, which have not previously been identified in individual scRNA-seq datasets, as well as a plethora of subpopulations of major cell types. Unlike previous studies, we find that adipocyte populations do share gene expression programs. We characterise compositional changes between lean and obese donors and between subcutaneous and visceral adipose tissue finding remodelling of the immune environment and changes in the balance between adipocytes and progenitor cells. We support the latter analysis by performing computational deconvolution of 1293 bulk RNA-sequencing samples from human subcutaneous and visceral adipose tissue depots. These analyses highlight that the WATLAS can be used for deconvolution with high accuracy and be used to discover compositional differences in obesity, type 2 diabetes, or other adipose tissue-related diseases.

In summary, JOINTLY is an advanced computational tool for characterising transcriptional states in sxRNA-seq data. By effectively addressing biological and technical variations, our method enables joint clustering and interpretation, facilitating the exploration of diverse biological processes across multiple datasets. To facilitate the use of JOINTLY, we have developed an R (www.github.com/madsen-lab/rJOINTLY) package, that integrates with common single-cell analysis frameworks, such as Seurat[13].

## Methods
### JOINTLY
The default workflow for the integration of datasets using JOINTLY involved six major steps:
1. Highly variable feature selection.
2. Normalisation and dimensional reduction.
3. Cell-cell similarity estimation using a kernel.
4. Optimising factors using hybrid linear and non-linear non-negative matrix factorisation.
5. Identification of gene modules.
6. Clustering and visualisation.

As a default, JOINTLY automatically selects (default = 1000) highly variable genes (HVGs) using a deviance-based measure[48,49] for each batch, and the union across all HVG sets is used for downstream analyses. Users can use Seurat for HVG identification or supply a user-defined list of HVGs.

In each batch, a size factor for each cell is calculated by dividing the sum of counts by 10,000 similarly to Seurat[13] or optionally using scran[50]. Each cell is then normalised and transformed using the logCPM method from Seurat[13] or optionally the shifted log transform[51]. The normalised and transformed counts are then standardised and decomposed into a reduced dimensional space using the selected HVGs. By default, JOINTLY uses consensus PCA for dimensional reduction. In consensus PCA, we calculate the variance-covariance matrix for each batch and calculate their sum weighted by the number of cells in each batch. This within-group variance-covariance matrix is decomposed using randomised singular value decomposition[52] into 20 components. In each batch, the amount of variance explained by these 15 components relative to a dataset-specific decomposition using the same number of components is evaluated. For batches where the common decomposition explains less than 80% of the variance explained by the dataset-specific decomposition, additional batch-specific components are added (see Supplementary Note 1 for a mathematical definition of consensus PCA).

The cell-cell similarity in each dataset is estimated using a data-adaptive alpha-decay kernel[9] based on the Euclidean distance between cells in the consensus PCA (or user-supplied) reduced dimensional space. The reduced dimensional space is also used to calculate a shared nearest neighbour graph using Seurat[13].

To optimise factors using hybrid linear and non-linear non-negative matrix factorisation, JOINTLY minimised the following loss function:

$$\arg\min_{F,H} = \alpha||\Phi(X_d) - \Phi(X_d)F_dH_d|| + \lambda\mathrm{Tr}\left(H_dL_dH_d^T\right)$$
$$+ \beta \sum_{d=1}^{D}\sum_{j\neq d}\left||V_jH_d - V_dH_d\right||\,\mathrm{s.t.}\,F_d \geq 0, H_d \geq 0, H_dH_d^T = I, d = 1,2,\ldots,D$$

(1)

This approach aims to reconstruct the normalised and standardised count matrix as well as the kernel space using the $H$ (clustering matrix), $F$ (basis matrix of the kernel), and $V$ (shared feature matrix) matrices. The first term in the loss function, weighted by $\alpha$, represents the reconstruction error within a single batch. Minimising this term encourages the clustering matrix to reconstruct the kernel. The second term, weighted by $\lambda$, applies graph regularisation to the $H$ matrix. Minimising this term encourages that cells close to each other in the consensus PCA space also are close to each other in the clustering matrix. The last term, weighted by $\beta$, represents the sum of differences in linear gene expression space across the clustering matrix between batches. Minimising this term encourages that the same genes contribute the same factors in the clustering matrix across batches. By default, the $H$ matrix is initialised per batch using fuzzy clustering in the consensus PCA space, and the $F$ and $V$ matrices using linear regression based on the $H$ matrix as well as the kernel and gene expression space, respectively. Next, the loss function is minimised using multiplicative updating algorithm[53] (see Supplementary Note 2 for a mathematical description of the algorithm) for 200 iterations. Finally, the $H$ matrices from each batch are concatenated and standardised per factor and then per cell.

To identify and score gene modules, the $V$ matrices are standardised per factor and then per gene for each batch and then averaged across batches. The averaged $V$ matrix is standardised per factor and then per gene. Genes are assigned to each factor by ordering the gene scores within each factor and finding the inflection point of the score distribution using the unit invariant knee method[54]. Genes with scores higher than the inflection point are assigned to the factor module and each factor module is scored in each cell using UCell[55].

For clustering and visualisation, the standardised $H$ matrix is used as input. For visualisation, UMAP coordinates were calculated using Seurat[13] and for clustering, we used hierarchical graph-based clustering[56], but any other clustering methods can be applied, such as Louvain clustering.

### Benchmarking
**Datasets.** Cell lines[11]: Counts were downloaded from Zenodo under [https://doi.org/10.5281/zenodo.3238275] and randomly split each cell type into two batches of equal size. We simulated a complex batch effect by introducing cell type-specific Poisson noise into the second batch. For each cell type, we drew a maximum noise level at random from a normal distribution with a mean and standard deviation of 0.5. Then we draw a noise fraction for each gene from a uniform distribution between zero and the maximum noise level for the cell type. Based on the average gene count across cells in the cell type and batch and the noise fraction, we calculate the mean of the noise distribution and randomly draw noise counts from the Poisson distribution. Adding gene-wise noise to gene-wise counts results in a new gene-wise Poisson distribution with a shifted mean corresponding to complex cell type- and gene-specific batch effects.

Lung[4]: Counts were downloaded from figshare under [https://doi.org/10.6084/m9.figshare.12420968.v8]. The dataset was subset to only contain cells from control patients retaining 10,046 cells and 15,148 genes across six batches. The donor ID was used as a batch

identifier, and cell type labels were transferred using a complementary human lung atlas[15].

Pancreas[19]: Counts from 20,784 cells and 30,378 genes across 12 donors were downloaded from NCBI Gene Expression Omnibus under accession code GSE114297. The donor ID was used as a batch identifier, and cell type labels were transferred using a complementary human pancreas atlas[24].

Kidney[20]: Counts were downloaded from cellxgene under accession code bcb61471-2a44-4d00-a0af-ff085512674c. The dataset was subset to only contain cells from healthy living donors retaining 20,497 cells and 27,305 genes across 18 batches. The donor ID was used as a batch identifier, and cell type labels were transferred using the single nucleus fraction from the same kidney atlas[20].

Liver[18]: Counts from 8444 cells and 32,922 genes across five donors were downloaded from cellxgene under accession code bd5230f4-cd76-4d35-9ee5-89b3e7475659. The donor ID was used as a batch identifier, and cell type labels were transferred using a complementary human liver atlas[22].

PBMC[21]: Counts were downloaded from cellxgene under accession code 03f821b4-87be-4ff4-b65a-b5fc00061da7. The dataset was subset to only contain cells from healthy adult donors retaining 46,993 cells and 33,193 genes across eleven batches. The donor ID was used as a batch identifier, and cell type labels were transferred using complementary human PBMC atlas[23].

**Label transfer**. For all datasets, except the cell line dataset, we transferred labels from similar public datasets as indicated above. Query and reference datasets were normalised to 10,000 UMI counts per cell and transformed using $\log(x+1)$. In the reference, differentially expressed (DE) genes were identified in each cell type versus all other cells with Scanpy[57] using Wilcoxon rank sum with log fold changes > 3. Gene sets are pruned, to remove genes that are in the first percentile highest DE in any other cell type. The top 50 genes are kept per cell type. Genes lists are further pruned for genes not expressed in the query dataset. We then performed two rounds of label transfer. In the first round, we trained a one-class support vector machine per cell type in the reference dataset using sklearn (linear kernel and nu = 0.5). Since probabilities in SVMs are uncalibrated, we identified classification optimal thresholds per cell type by maximising the geometric mean between the sensitivity and the specificity on the reference data and used these to label cells in the query dataset. Cells labelled as multiple cell types were labelled as unknown, as were cells not labelled as any cell type. In the second round of labelling, we used the labelled query cells to train a linear support vector classifier using sklearn (maximum of 1000 iterations with balanced class weights). Again, we identified classification optimal thresholds per cell type by maximising the geometric mean between the sensitivity and the specificity and obtained the final labels for all cells in the query dataset, and unlabelled or ambiguously labelled cells were removed from the dataset.

**Integration parameters**. JOINTLY was run with default parameters on all datasets, except for the cell line dataset, which was run using 15 components.

scVI[3] parameters were obtained from online tutorials:[58] The dataset was subset to highly variable genes selected by JOINTLY and integrated based on raw counts running 500 epochs using early stopping with patience of 10. The integration was evaluated and visualised using 10 dimensions.

Harmony[2] parameters were obtained from online tutorials:[59] The dataset was normalised, scaled, and dimensionally reduced using Seurat based on highly variable genes selected by JOINTLY. Harmony was run using default parameters and evaluated and visualised using 20 dimensions.

Scanorama[12] parameters were obtained from online tutorials:[60] The dataset was normalised using scanpy[57] and subset to highly variable genes selected by JOINTLY. Each batch was scaled independently and integrated. The integration was evaluated and visualised using 50 dimensions.

LIGER[7] parameters were obtained from online tutorials:[61] The dataset was normalised, scaled (per batch, without centring), and dimensionally reduced using Seurat based on highly variable genes selected by JOINTLY. The datasets were integrated using 20 factors and a lambda of five following my quantile normalisation. The integration was evaluated and visualised using 20 components.

fastMNN[1] parameters were obtained from online tutorials:[62] The datasets were integrated using auto merge and highly variable genes selected by JOINTLY. The integration was evaluated and visualised using 20 dimensions.

Seurat[13] parameters were obtained from online tutorials:[63] The dataset was normalised, scaled, and dimensionally reduced using Seurat based on highly variable genes selected by JOINTLY. Integration anchors were identified using reciprocal PCA and highly variable genes selected by JOINTLY. The datasets were integrated using default parameters, except for datasets with small batches (less than 100 cells) where a k.weight of 50 was used. The integration was evaluated using 30 dimensions.

scGPT[14] parameters were obtained from online tutorials:[64] The dataset was normalised and scaled using Scanpy based on highly variable genes selected by JOINTLY. The datasets were integrated using the parameters supplied in the tutorial, except that cell type labels were not used for train test splitting making the method unsupervised. The integration was evaluated using 512 dimensions.

**Integration evaluation**. We evaluated how well clustering could recover cell types (using only cell types with at least 10 cells) by clustering each dataset based on the integrated reduced dimensional space using hierarchical graph-based clustering[56] into between 2 and 50 clusters. For each clustering solution, we calculated the adjusted rand index (ARI) between the clusters and the transferred labels for the entire dataset and each batch. A higher value indicates better agreement between clusters and cell type labels.

We evaluated how well cell types and batches mix using the local inverse Simpson's Index (LISI)[2], which is a metric that measures the distributions of categorical variables over local neighbourhoods. We calculated LISI for cell type labels and batch labels and rescaled the metrics as previously described[4] such that a value of 0 corresponds to low cell type separation and low batch integration, whereas a value of 1 corresponds to high cell type separation and high batch integration,

Finally, we evaluated the distance between cell types and batches using the average silhouette width (ASW). The silhouette width measures the relationship between within-group distances and between-group distances. We calculated scores for cell types and batches as previously described[4], where scores are transformed such that values of 0 correspond to worst performance (low cell type separation and high batch separation), whereas values of 1 correspond to best performance (high cell type separation and low batch separation).

To account for the stochasticity of some of the methods, we ran all methods five times and reported across all metrics for the integration run with the highest overall ARI. Ranks for each metric and dataset was calculated using two significant digits and setting the minimum rank for ties. The overall rank across tissues was calculated by averaging across the ranks of individual metrics and setting the minimum rank for ties.

**Neighbourhood conservation**. To evaluate neighbourhood conservation between two datasets, we calculated the 100 nearest neighbours for each cell in each dataset based on the dataset-specific reduced dimensional space from JOINTLY, and for cells represented in both datasets, we summed the number of shared neighbours. This

total of shared neighbours is divided by the total number of possible shared neighbours.

## Over-correction analysis

**Dataset, integration, and evaluation.** Datasets were merged pairwise to generate all possible combinations of the 5 datasets used for benchmarking. To merge datasets, the datasets were subset to genes detected in both datasets and combined.

The combined datasets were integrated as described under 'Benchmarking' with the exception that JOINTLY was run using PCA rather than consensus PCA. To evaluate integration, we calculated an over-correction score by calculating the integration LISI using the tissue as the label. Next, we split each dataset into each tissue and calculated the integration and cell type LISI within each tissue. Ranks for each metric were calculated using three significant digits and setting the minimum rank for ties. The overall rank was calculated by averaging across the overall rank within each tissue separately and the over-correction rank and setting the minimum rank for ties.

## Differential expression analysis

To identify differentially expressed genes, we created pseudo-bulk expression levels per donor for each cell type in each tissue. Next, we prefiltered genes that did not have at least 10 counts across all samples We performed pairwise differential expression between all cell types. To identify shared endothelial cell marker genes, we extracted all pairwise tests between endothelial cells from the Lung or the Pancreas dataset and any non-endothelial cell type. We combined the *p*-values from all tests using the harmonic mean and FDR-corrected the combined *p*-value. Finally, shared marker genes were defined as genes with a combined FDR ≤ 0.01 and log2 fold change ≥ 1.5 in all pairwise tests. Endothelial state markers were defined as genes with an FDR ≤ 0.01 and absolute log2 fold change ≥ 1.5 between Lung- and Pancreas-derived endothelial cells.

**Pathway and transcription factor identification.** Enrichment analysis of gene modules was performed using enrichR[65] with default parameters using the Reactome_2022 databases for pathway identification and ChEA_2022 for transcription factor identification. *P*-values were corrected using FDR, and terms with an FDR-corrected *P*-value less than 0.05 were kept.

## Interpretability analysis

**Datasets.** Emont[25]: Counts were downloaded from the Single Cell Portal under accession code SCP1376. The dataset was subset to total visceral adipose tissue samples, containing 80,0085 cells across 10 donors.

Mouse gastrulation[28]: Counts were downloaded using scVelo[27] and contain 9815 cells across three sequencing batches.

**Cell type labelling.** Enrichment analysis of gene modules was performed using enrichR[65] with default parameters using the Azimuth_Cell_Types_2021 and CellMarker_Augmented_2021 databases. In the Azimuth_Cell_Types_2021, which does not contain categories specifically from adipose tissue, all terms associated with Neurons were removed, as well as terms without cell type or tissue names. In the CellMarker_Augment_2021, which does contain categories specifically associated with adipose tissue, we only retained categories from adipose tissue, excluding brown and beige adipose tissue. All terms with a false discovery rate corrected *P*-value less than 0.1 were kept and for each tested factor, the top three terms ordered by the combined score[65] were reported.

**Pathway identification.** Enrichment analysis was performed using enrichR[65] with default parameters using the WikiPathway_2021_Human,

KEGG_2021_Human, and Reactome_2022 databases for the human adipose tissue and the WikiPathways_2019_Mouse database for the mouse gastrulation dataset. *P*-values were corrected using FDR, terms with an FDR-corrected *P*-value less than 0.05 were kept and selected pathways were shown.

## Constructing a white adipose tissue atlas (WATLAS)

**Datasets.** Tabula Sapiens[15]: Counts were download from Figshare under https://doi.org/10.6084/m9.figshare.14267219[66] and cells from the adipose tissue from two donors were extracted. Subsequently, cells with an abnormal relationship between the number of detected genes and the total counts were removed in two rounds by fitting a linear regression model to the two variables and removing cells with an absolute residual above 2 in the first round and above 0.8 in the second round. Finally, cells with a fraction of counts derived from mitochondrial genes above 0.2 were removed.

Jaitin[31]: Counts from one donor were downloaded from the NCBI Gene Expression Omnibus under accession code GSE128518 and metadata from Bitbucket[67] repository amitlab/adipose-tissue-immune-cells-2019 [https://bitbucket.org/amitlab/adipose-tissue-immune-cells-2019/src/master/]. Cells passing quality control by the original authors (as indicated in the metadata) were kept and further filtered to remove cells with an abnormal relationship between the number of detected genes and the total counts were removed by fitting a linear regression model to the two variables and removing cells with an absolute residual above 0.8 for run 22 and 0.7 for run 55. Finally, cells with a fraction of counts derived from mitochondrial genes above 0.15 were removed.

Vijay[30]: Counts were downloaded from the NCBI Gene Expression Omnibus under accession code GSE129363. Cells with a log2-transformed total count and a total number of features higher than six were kept. Next, cells with an abnormal relationship between the number of detected genes and the total counts were removed by fitting a linear regression model to the two variables and removing cells with an absolute residual above 1.0. Finally, cells with a fraction of counts derived from mitochondrial genes above 0.2 were removed, and the two diabetic donors were removed.

Hildreth[29]: Counts from six donors were downloaded from the NCBI Gene Expression Omnibus under accession code GSE155960. All datasets were merged and cells with an abnormal relationship between the number of detected genes and the total counts were removed by fitting a linear regression model to the two variables and removing cells with an absolute residual above 0.8. Finally, cells less than 500 counts or a fraction of counts derived from mitochondrial genes above 0.15 were removed.

Emont[25]: Counts were downloaded from the Single Cell Portal under accession code SCP1376. The dataset was split into two independent sets containing samples from the stromal vascular fraction and whole adipose tissue, respectively.

Barboza[32] We obtained a dataset from whole adipose tissue from the original authors.

**Integration.** After processing and filtering, each dataset was processed with JOINTLY using 15 components, clustered using hierarchical graph-based clustering[56] into coarse cell types, and labelled based on marker genes. Clusters, that could not be assigned to specific cell types, had high mitochondrial content, or low specificity of marker genes were removed. We concatenated the datasets, harmonising metadata, and gene names, retaining only genes with a total count above 10 and expressed in at least one cell. We integrated the datasets using scANVI[33] using 15 latent dimensions and parameters from scArches[68]. The cells were clustered and separated into fine cell types and states, as well as removing a few clusters containing low-quality cells (8097 cells removed, 2.6%). The final dataset was re-integrated using scANVI and the fine-grained labels.

**Composition analysis.** Cell type or state fractions were calculated for each donor and modelled using linear regression correcting for variables as indicated in the figures. Marginalised effect sizes and confidence intervals were calculated using emmeans.

**Cell type decomposition.** GTEx adipose bulk RNA-seq data were deconvoluted using bisque[69]. Pseudo-bulk references were generated for both VAT and SAT depots of the WATLAS at intermediary label granularity. Marker genes for decomposition were identified using presto[70] with an AUC threshold of 0.6. After deconvolution, we removed mesothelial cells from VAT and rescaled each bulk sample to sum to one. We tested for association between proportions and categorial variables using linear regression and estimated effect sizes and confidence intervals using estimated marginal means.

**Differential expression analysis of adipocyte populations.** To identify differentially expressed genes between adipocyte populations, or between depots within adipocyte populations, we created pseudo-bulk expression levels per donor in the Emont et al. and Barboza et al. datasets by summarising counts for all genes. Next, we prefiltered genes that were not expressed in at least 20 cells in either study as well as genes that were identified by edgeR as lowly expressed in the pseudo-bulk count matrix from either study. We performed differential expression analysis using edgeR correcting for study (and depot in the case of comparing adipocyte populations). We defined depot-specific genes as differentially expressed genes (FDR ≤ 0.05 and an absolute log2 fold change ≥ 1.5) between VAT and SAT in any one adipocyte subpopulation, and defined subpopulation marker genes as differentially expressed genes (FDR ≤ 0.05 and log2 fold change ≥ 1.5) in one-versus-all comparisons as well as log2 fold change ≥ 1.3 across all one-versus-one comparisons. For visualisation of depot- and subpopulation-specific marker genes we used limma to batch-correct the normalised counts.

## Statistics and reproducibility

The statistical analysis of the data was described in the above methods and can be reproduced from scripts available on GitHub (http://www.github.com/madsen-lab/JOINTLY_reproducibility). No statistical method was used to predetermine sample size. Benchmarking datasets were subset to only include cells from healthy controls in applicable datasets as described in the above methods to ensure that within datasets the major axis of variation could be assumed to be shared between batches. Data included in the WATLAS was rigorously quality controlled as described in the above sections to exclude low quality cells. The experiments were not randomised. The investigators were not blinded to allocation during experiments and outcome assessment.

## Reporting summary

Further information on research design is available in the Nature Portfolio Reporting Summary linked to this article.

## Data availability

All relevant data supporting the key findings of this study are available within the article and its Supplementary Information files. The dataset containing mixtures of cell lines is available through Zenodo under [https://doi.org/10.5281/zenodo.3238275]. The datasets used for benchmarking is available through figshare under [https://doi.org/10.6084/m9.figshare.12420968], cellxgene under accession codes bcb61471-2a44-4d00-a0af-ff085512674c, bd5230f4-cd76-4d35-9ee5-89b3e7475659, and 03f821b4-87be-4ff4-b65a-b5fc00061da7, and NCBI Gene Expression Omnibus under accession code GSE114297. The datasets used for investigating interpretable factors and for building WATLAS are available from the NCBI Gene Expression Omnibus under accession codes GSE128518, GSE129363, and GSE155960 and the Single Cell Portal under accession code SCP1376. The processed data for the white adipose tissue atlas is explorable at the Single Cell Portal[34] under accession code SCP2289. The model weights for transfer learning and integrating new datasets are available at Zenodo[35] under https://doi.org/10.5281/zeodo.8086433 [https://zenodo.org/records/8086433]. The processed datasets used for evaluation and the embeddings, results, and summaries are available at Zenodo[71] under https://doi.org/10.5281/zenodo.8434958 [https://zenodo.org/records/8434958]. Source data are provided with this paper.

## Code availability

JOINTLY is available as an R package on GitHub (http://www.github.com/madsen-lab/rJOINTLY) and Zenodo[72] under https://doi.org/10.5281/zenodo.10159672. Scripts for reproducibility are available on GitHub (http://www.github.com/madsen-lab/JOINTLY_reproducibility).

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

## Acknowledgements

This work was supported by grants from the Novo Nordisk Foundation (NNF21SA0072102, J.G.S.M. and NNF21OC0068929, J.G.S.M.), as well as the Danish National Research Foundation (DNRF141, J.G.S.M.) to the Center for Functional Genomics and Tissue Plasticity (ATLAS), Sino-Danish Center for Education and Research (A.F.M.), and Aage og Johanne Louis-Hansen Foundation (J.nr.20-2B-6705, A.F.M.). Computation for this project was performed using the UCloud interactive HPC system, which is managed by the eScience Center at the University of Southern Denmark. We thank Professor Carey Lumeng from the University of Michigan for sharing unpublished data and thank all the members of the Madsen group for fruitful discussions that improved the manuscript.

## Author contributions

A.F.M. and J.G.S.M. conceptualized the method. A.F.M. derived the optimisation algorithm. A.F.M and J.G.S.M implemented the code, performed analysis, and prepared figures. J.G.S.M. supervised the work. J.G.S.M. wrote the manuscript with inputs from A.F.M. All authors contributed to its editing.

## Competing interests

The authors declare no competing interests.
