## [Peer Review File · Nature Communications]

JOINTLY: Interpretable joint clustering of single-cell transcriptomesReviewer #1 (Remarks to the Author):

In this manuscript, Andreas et al. reported a new method JOINTLY which removes batches in data integration by jointing clustering of individual scRNA-seq datasets. Compared with the existing data-integration approaches, the authors highlighted this new method preserved the subtle cell states across different biological conditions when performing batch correction. This kind of topics has been a hotspot in the single-cell field and will be of interest to the community. But many data-integration approaches have already been documented and some recent ones showed good performance in preserving cell states, so the authors might need to provide strong evidence to show outperformance or novelty of JOINTLY, especially JOINTLY is based on PCA and NMF which are widely used in the existing approaches.

Major comments:

(1) In the method "consensus PCA" (supplementary note 1), the authors provided the principal idea of formula derivation from individual V_d to within-group V , however, the details are not clear, the authors did not show the shape of matrix X_d , V_d , or V , even U .

Further, in the section "JOINTLY" (Online Methods), the authors interpreted that "Cell-cell similarity in each dataset ... in the consensus PCA (or user-supplied) reduced dimensional space.", this description will easily mislead the audience that V is cell x cell matrix. However, if X_d is gene-cell matrix ($N \times M$ with N genes and M cells), then V_d is $M \times M$ matrix (from $X_d^T * X_d$). It is difficult to get V by summing multiple V_d together when their cell numbers are different, and U , U^T , and V_g are "non-conformable" in matrix multiplication. In the other scenario (I thought this should be the correct one after I read LIGER method details¹), V is gene x gene matrix ($N \times N$), then some principles of JOINTLY may not be appropriate in the manuscript. Maybe, R code of JOINTLY still works, but the authors' interpretation in the method part made the mathematical model of JOINTLY defective, e.g. in supplementary note 2, "The required inputs are kernel matrix ... in consensus PCA space ... an adjacency and a degree matrix (cell by cell) both of which are calculated from a shared nearest neighborhood graph built on consensus PCA." The authors may need to polish the method part and make sure JOINTLY has a rigorous framework. But if my understanding is not correct, the authors can provide more details.

(2) In the method "graph regularized kernel NMF", the kernel matrix K , the degree matrix D and the adjacency matrix A make direct impact on H and F which determine the final results, and K , D and A are all based on the "consensus PCA" that is calculated from the within-group variance V of specific/input datasets and will be changed when containing different datasets. In other words, even integrating scRNA-seq datasets of the related samples, the JOINTLY results might be different, depending on which datasets the users input. The authors may investigate the consistency of the JOINTLY by integrating a series of related datasets, estimating whether the results are consistent when including all the datasets and just part of them. The authors can use the datasets with time axis, such as embryoid body scRNA-seq datasets from Kevin et al.².

(3) The authors may perform benchmarking tests, both cell-type and batch, for all the JOINTLY results; otherwise, the audience have to measure its performance by vision. To develop a reliable analytic tool, quantitative measurement of JOINTLY for handling complicated datasets might be necessary in this study.

(4) One of the major advantages of JOINTLY is "JOINTLY also generates interpretable factors, ..., and discover active biological processes". However, in the benchmarking tests (Fig. 2), JOINTLY did not show outperformance but similar cell-type scores to mainstream state-of-the-art tools. Therefore, in the section "JOINTLY retains biological variation across conditions", the authors may need to display the outperformance of JOINTLY in retaining biological variation globally and quantitatively, not just discussing a part of cell types.

(5) The authors showed JOINTLY's interpretable factors in several sections of this manuscript, but most of them are for cell types with unique gene markers. I am curious that whether JOINTLY's this character can also be applied to the datasets with transition state cells which may not have unique markers, as transition state cells are also very important contents in biological events.

(6) The application of JOINTLY is another important question. For my understanding, JOINTLY itself does not have a function to do gene expression alignment across conditions. This makes JOINTLY difficult to explore novel cell-type markers or differentially expressed (DE) genes. In this case, JOINTLY can be easily replaced by other mainstream data-integration tools when they find similar cell types. The authors showed JOINTLY application in a white adipose tissue atlas by guiding cell types in data integration using scANVI (Fig. 5), but they did not provide the necessity

of JOINTLY in the process. If the authors can show outperformance of JOINTLY + scANVI when compared with other integration tools, the conclusion might be more persuasive.

Minor comments:

(1) The authors may notify that JOINTLY uses clustering results of other cluster methods in the "Introduction" section. Some interpretations, such as "JOINTLY achieves state-of-the-art joint clustering performance, but unlike existing methods with comparable clustering performance, JOINTLY also generates ...", may mislead the audience that JOINTLY has its own clustering method.

(2) Typo "1.300", "~10.000", ...

(3) Typo "in addition to joint clustering ..."

(4) The audience cannot know the quantitative difference in Fig. 2B and 2C.

(5) The statement of JOINTLY's "interpretable" may not be accurate. JOINTLY, LIGER and other NMF-based utilize cell-type-specific gene modules to decompose and integrate datasets, while fastMNN, Scanorama, Seurat, and et al use shared cell types, the cell-type-specific gene modules can also be generated based on cell types after integration.

(6) In differential expression analysis of the pseudo-bulk counts, the authors did not show how to correct batch effects across datasets and how to utilize pseudo-bulk analysis to represent transcriptomic characters of single cell counts. Because this is a complicated process³, the authors may provide more details.

(7) Fig. 4F and 5H are not in the text.

Reference:

1. Welch JD, Kozareva V, Ferreira A, Vanderburg C, Martin C, Macosko EZ. Single-Cell Multi-omic Integration Compares and Contrasts Features of Brain Cell Identity. *Cell*. 2019 Jun 13;177(7):1873-1887.e17. doi: 10.1016/j.cell.2019.05.006. Epub 2019 Jun 6. PMID: 31178122; PMCID: PMC6716797.

2. Moon KR, van Dijk D, Wang Z, Gigante S, Burkhardt DB, Chen WS, Yim K, Elzen AVD, Hirn MJ, Coifman RR, Ivanova NB, Wolf G, Krishnaswamy S. Visualizing structure and transitions in high-dimensional biological data. *Nat Biotechnol*. 2019 Dec;37(12):1482-1492. doi: 10.1038/s41587-019-0336-3. Epub 2019 Dec 3. Erratum in: *Nat Biotechnol*. 2020 Jan;38(1):108. PMID: 31796933; PMCID: PMC7073148.

3. Squair JW, Gautier M, Kathe C, Anderson MA, James ND, Hutson TH, Hudelle R, Qaiser T, Matson KJE, Barraud Q, Levine AJ, La Manno G, Skinnider MA, Courtine G. Confronting false discoveries in single-cell differential expression. *Nat Commun*. 2021 Sep 28;12(1):5692. doi: 10.1038/s41467-021-25960-2. PMID: 34584091; PMCID: PMC8479118.

Reviewer #2 (Remarks to the Author):

The main topic of this manuscript is the development of an algorithm called JOINTLY for joint clustering of single-cell and single-nucleus RNA-sequencing datasets across batches. The algorithm uses a standardised H matrix as input for clustering and visualisation, and the V matrices are used to identify and score gene modules. The genes are assigned to each factor module by ordering the gene scores within each factor and finding the inflection point of the score distribution using the unit invariant knee method. Finally, the factor modules are scored in each cell using UCell. The paper demonstrates how JOINTLY can be used to create a tissue atlas by clustering and labelling cell types and states in white adipose tissue from 6 different studies. Based on these labels, the authors create a reference atlas of white adipose tissue (WAT-LAS) deeply characterises the transcriptome of 43 cell types and states.

While the paper is clearly written, it would benefit from a more specific description of the key "innovation". For example, what specific aspect of interpretability or what is the actual gap it is trying to address. Some comments are provided below:

1. The authors have effectively utilized matrix decomposition as a foundational element in their method. While this approach provides enhanced robustness, it could benefit from further

exploration to match the capabilities exhibited by contemporary deep learning-based alternatives. Additionally, enhancing the evaluation process through a comprehensive comparison with modern deep learning-based competitors would enrich the overall analysis.

2. While the title mentioned "interpretable clustering," the authors have mainly focused on the challenge of batch effect elimination and mitigating the risk of over-correction. Thus the comparison study will benefit from comparison with data integration approaches mentioned in "Benchmarking atlas-level data integration in single-cell genomics" (<https://www.nature.com/articles/s41592-021-01336-8>). Following from point 1, possibly calculating statistics and metrics described in the benchmarking paper so that direct comparison can be made without re-running all the methods.

3. In the abstract, the statement, "Unlike existing state-of-the-art methods, JOINTLY is interpretable,..." is an overclaim. Numerous interpretable clustering methods already exist, and many techniques permit post-hoc processing or DE analysis to enhance interpretability. So exactly what is the additional "interpretation" one gains from JOINTLY? In the introduction, the authors referenced LIGER, but there are other clustering methods specifically designed for single-cell data, such as <https://academic.oup.com/bib/article/24/4/bbad199/7176310> that claim to be interpretable. It is vital to succinctly articulate the unique differences and innovations this paper offers compared to the current literature.

4. Currently, I believe it's important to recognize that both supervised and unsupervised clustering methods aim to identify "cell types". It's important that in the evaluation the focus is not "narrow" towards "cell type discovery only". One should also consider how JOINTLY performed with methods that incorporate biological knowledge.

5. Minor: The clustering results that the authors have presented are satisfactory, but still have the potential for further refinement to achieve a cutting-edge advancement in the field. For example, the ARI results shown in Fig2A of JOINTLY and Seurat (common approach) are both 0.9. Is there datasets or situation where JOINTLY is significantly improved from Seurat?

6. Minor: There are several typos that need to be corrected. For example, at the bottom in page 12, "Minimizing this term encourages the that the same genes contribute the same factors in the clustering matrix across batches."

Code and Software Review

1. Data and source code availability and replicability of the results in the paper. The GitHub code has been thoughtfully organised, incorporating reasonable documentation and utilising optimised C++ functions for enhanced efficiency during matrix decomposition. That being said, we have a number of concerns:

(a) The accessibility of the GitHub link provided in the manuscript for reproducibility purposes (i.e. http://www.github.com/madsen-lab/JOINTLY_reproducibility). Unfortunately, attempting to access the link results in a 404 Error, thereby preventing us from fully evaluating the reproducibility of the paper's results.

(b) While the well-structured R code and detailed documentation are commendable aspects, the inability to access the private GitHub repository raises significant obstacles in verifying and replicating the research findings.

(c) The program lacks explicit documentation for all dependencies, specifying compatible versions that users can install seamlessly. In this case, with the "SharedObject" package (we observed the "package 'SharedObject' is not available for BioConductor version '3.17'" error when trying to install rJOINTLY). This scenario might require users to invest considerable time in troubleshooting and resolving version conflicts or, in some cases, necessitate the creation of a separate R environment solely for running the "rJOINTLY" package.

2. Ability to run the tool successfully. Due to insufficient documentation regarding compatible versions of dependencies, package installation was challenging and time consuming. This obstacle made it cumbersome to set up the necessary environment for running the tool. A possible solution is to provide a CodeOcean link with an installed R environment. On the other hand, the test data, i.e., the file `"../data/data_seurat.rds"` mentioned in README is not provided, so we used our own dataset for testing. The main `"jointly"` function could be successfully executed in our test.

3. Code documentation for following the algorithm. The code has been thoughtfully organized and reasonably documented. It is noteworthy that the code utilized C++ scripts, which would be a little bit difficult for users who primarily use R. In addition, a more detailed tutorial will increase usability.

4. Software to be run on a widely available operating system. We encountered many challenges in package installation, which was mainly caused by version conflicts of dependencies. Once the conflicts are handled, there would be no reason for the program to be successfully run on a widely available operation system.

Reviewer #3 (Remarks to the Author):

I co-reviewed this manuscript with one of the reviewers who provided the listed reports as part of the Nature Communications initiative to facilitate training in peer review and appropriate recognition for co-reviewers.

Point-by-point response to reviewers

Reviewer #1:

In this manuscript, Andreas et al. reported a new method JOINTLY which removes batches in data integration by jointing clustering of individual scRNA-seq datasets. Compared with the existing data-integration approaches, the authors highlighted this new method preserved the subtle cell states across different biological conditions when performing batch correction. This kind of topics has been a hotspot in the single-cell field and will be of interest to the community. But many data-integration approaches have already been documented and some recent ones showed good performance in preserving cell states, so the authors might need to provide strong evidence to show outperformance or novelty of JOINTLY, especially JOINTLY is based on PCA and NMF which are widely used in the existing approaches.

RESPONSE: We thank the reviewer for recognizing the importance of preserving cell states across biological conditions. Please find below and point-by-point response to major comments and minor comments.

Major comments:

Point 1:

In the method “consensus PCA” (supplementary note 1), the authors provided the principal idea of formula derivation from individual V_d to within-group V , however, the details are not clear, the authors did not show the shape of matrix X_d , V_d , or V , even U .

Further, in the section “JOINTLY” (Online Methods), the authors interpreted that “Cell-cell similarity in each dataset ... in the consensus PCA (or user-supplied) reduced dimensional space.”, this description will easily mislead the audience that V is cell x cell matrix. However, if X_d is gene-cell matrix ($N \times M$ with N genes and M cells), then V_d is $M \times M$ matrix (from $X_d^T * X_d$). It is difficult to get V by summing multiple V_d together when their cell numbers are different, and U , U^T , and V_g are “non-conformable” in matrix multiplication. In the other scenario (I thought this should be the correct one after I read LIGER method details1), V is gene x gene matrix ($N \times N$), then some principles of JOINTLY may not be appropriate in the manuscript. Maybe, R code of JOINTLY still works, but the authors’ interpretation in the method part made the mathematical model of JOINTLY defective, e.g. in supplementary note 2, “The required inputs are kernel matrix ... in consensus PCA space ... an adjacency and a degree matrix (cell by cell) both of which are calculated from a shared nearest neighborhood graph built on consensus PCA.” The authors may need to polish the method part and make sure JOINTLY has a rigorous framework. But if my understanding is not correct, the authors can provide more details.

RESPONSE: We thank the reviewer for pointing this confusion out and understand the concern regarding matrix shapes but assure the reviewer that the framework is rigorous. We believe that part of the confusion can be attributed to overlapping conventions of variable naming. Originally, we used V to describe variance-covariance matrices (gene-by-gene, $m \times m$) computed in CPCA and in the JOINTLY decomposition for an interpretable gene-by-factor matrix (gene by factor, $m \times f$). We have clarified the shape of the individual matrices in Supplementary Note 1 (**NEW TEXT**), as well as changed the naming convention of variance-covariance matrices to C to avoid confusion.

To summarize here, in dataset d with n_d cells and m highly variable genes, the matrix X_d is of dimensions $m \times n_d$. The matrix multiplication $X_d * X_d^T$ results in a matrix C_d ($m \times m$), which is a gene-gene co-expression matrix.

Weighted summation of C_d across all datasets d results in the matrix C_g gives us the within-group variance-covariance matrix ($m \times m$).

We use a QR decomposition-based approximation of compact SVD called randomized SVD to decompose the C_g matrix into k factors such that $C_g = U * \Sigma * V^T$ where C_g is the within-group variance-covariance matrix ($m \times m$), the diagonal of Σ represents the singular values ($k \times k$), U is the left-singular vectors ($m \times k$) and V is the right-singular vectors ($k \times m$).

We can then use the left-singular vectors to derive the reduced dimensional space for each dataset d by $U^T * X_d^T$, which results in a matrix E_d that is factor by cell ($k \times n_d$). For each dataset, we calculate the proportion of variance explained by this common decomposition using E_d . For datasets with less than 80% (default) variance explained, we factor out the common left-singular vectors from the dataset-specific C_d matrix ($m \times m$) by $C_d - U * U^T * C_d$ and repeat the randomized SVD to identify dataset-specific left-singular vectors, U_d , explaining residual variance.

We concatenate the common left-singular vectors with any dataset-specific vectors $U_{all} = (U, U_d)$ which is a $m \times k_{all}$ sized matrix, where k_{all} is equal to the initially chosen k and any additional dataset-specific vectors. Finally, each dataset into is decomposed into a reduced dimensional space using $U_{all}^T * X_d^T$ resulting in the final embedding matrix E_d ($k_{all} \times n_d$). These matrices are then used for cell-cell similarity estimation using kernel-based methods resulting in a kernel matrix K_d with dimensions $n_d \times n_d$ as well as a graph-based similarity using shared nearest neighbors resulting in matrix A_d , which is also $n_d \times n_d$. The matrix K_d , A_d and X_d (the expression value) are used for decomposition in JOINTLY to estimate matrices H_d ($f \times n_d$), F_d ($n_d \times f$) and V_d ($m \times f$) where f is the chosen rank in JOINTLY. The H_d matrix is used for embedding and clustering and the V_d matrix is used for interpretation.

Point 2:

In the method “graph regularized kernel NMF”, the kernel matrix K , the degree matrix D and the adjacency matrix A make direct impact on H and F which determine the final results and K , D and A are all based on the “consensus PCA” that is calculated from the within-group variance V of specific/input datasets and will be changed when containing different datasets. In other words, even integrating scRNA-seq datasets of the related samples, the JOINTLY results might be different, depending on which datasets the users input. The authors may investigate the consistency of the JOINTLY by integrating a series of related datasets, estimating whether the results are consistent when including all the datasets and just part of them. The authors can use the datasets with time axis, such as embryoid body scRNA-seq datasets from Kevin et al.

RESPONSE: In all integration methods, the integration is dependent on the number and nature of datasets that are supplied to the algorithm. For some methods, such as FastMNN, the result even depends on the order of the datasets. Specifically, for JOINTLY, we agree with the reviewer that the consensus PCA is a key step where datasets impact each other in the embedding as they all contribute to the U_{all} matrix (as detailed additionally above). The idea behind this is to ensure that our algorithm prioritizes sources of shared variance before considering dataset-specific variances. We have tested the impact of using different datasets and added a paragraph to the manuscript on (**NEW TEXT** page 4) describing these new analyses:

One of the differences between JOINTLY and other methods is that JOINTLY uses consensus PCA (see Methods) as a basis for integrating samples. Therefore, the initial decomposition of each sample depends on the other samples. Thus, integration performance is dependent on the number and similarity of input samples. To evaluate the extent of this dependency, we removed all sample combinations from the human liver dataset

resulting in 25 subsamples with two to four batches. We found no difference in the ARI, in the average cell type and integration LISI nor the cell type and batch ASW for subsamples with two to four batches compared to the full dataset with five batches (**NEW ANALYSIS** Figure 2D-H), although we did find that with fewer batches, the standard error of the mean increases, indicating that integration performance becomes more variable. We also evaluated how well the nearest neighbours are conserved between each sub-sample and the full dataset and found that approximately 50 – 60% of the nearest neighbours are conserved with a minor decrease with fewer batches, which is similar in range to the variation between different integrations on the full dataset (**NEW ANALYSIS** Figure 2I). Collectively, this indicates that the performance of JOINTLY is not strongly dependent on the number of input datasets.

Point 3:

The authors may perform benchmarking tests, both cell-type and batch, for all the JOINTLY results; otherwise, the audience have to measure its performance by vision. To develop a reliable analytic tool, quantitative measurement of JOINTLY for handling complicated datasets might be necessary in this study.

RESPONSE: In the manuscript there are two main figures that relate to benchmarking tests for cell type and batches:

- In figure 2, we compare the performance of JOINTLY to other methods in terms of clustering, separation and mixing of cell types and separation and mixing of batches. All of these were assessed quantitatively using metrics commonly used in the field (Luecken et al. (2022): Benchmarking atlas-level data integration in single-cell genomics, Nature Methods). The quantitative metrics are summarized as ranks in the main figure, as a deluge of numbers are not visually appealing or particularly easy to decode. However, for the interested reader, we have provided all quantitative values in Supplementary Table S1.
- In figure 3, we compare JOINTLY to other methods in terms of mixing tissues (over-correction), as well as mixing cell types and batches within tissues. We have added a new figure with the ranks for each method, as well as an overall rank (**NEW ANALYSIS** Figure 3A). In addition, we have also provided these quantitative values in Supplementary Table S2.

Point 4:

One of the major advantages of JOINTLY is “JOINTLY also generates interpretable factors, ..., and discover active biological processes”. However, in the benchmarking tests (Fig. 2), JOINTLY did not show outperformance but similar cell-type scores to mainstream state-of-the-art tools. Therefore, in the section “JOINTLY retains biological variation across conditions”, the authors may need to display the outperformance of JOINTLY in retaining biological variation globally and quantitatively, not just discussing a part of cell types.

RESPONSE: We agree with the reviewer that JOINTLY performs on par with existing mainstream state-of-the-art methods, but has additional benefits, such as interpretability and more robustness against over-correction. In figure 3, we delved into endothelial cells because they represent a glaring example of where other contemporary tools fail and overcorrect. We have added a new data to the manuscript on the quantitative measures of the extend of which JOINTLY and other methods to mixes tissues (over-correction), as well as how well they mix cell types and batches within tissues across the entire dataset. We have added a new figure with the ranks for each method, as well as an overall rank (**NEW ANALYSIS** Figure 3A). In addition, we have also provided the quantitative values in Supplementary Table S2.

Point 5:

The authors showed JOINTLY's interpretable factors in several sections of this manuscript, but most of them are for cell types with unique gene markers. I am curious that whether JOINTLY's this character can also be applied to the datasets with transition state cells which may not have unique markers, as transition state cells are also very important contents in biological events.

RESPONSE: The reviewer raises an important point, which we had not considered in our initial benchmark, and we agree that analysis of continuous biological processes, such as differentiation is an important use-case of JOINTLY. To address this question, we analyzed a dataset containing three batches of mouse cells undergoing erythropoiesis using JOINTLY and scVelo (Bergen et al. (2020): Generalizing RNA velocity to transient cell states through dynamical modeling, Nature Biotechnology), which is a state-of-the-art method for the estimation of latent time. We found that JOINTLY does a good job at capturing cells in order of latent time and that the interpretable factors of JOINTLY can be used to gain insight into the biological processes ongoing during differentiation. We added a paragraph to the manuscript (**NEW TEXT** page 6) describing these new analyses:

In addition to discrete cell types, scRNA-seq can also be used to probe continuous biological processes, such as differentiation. To test JOINTLY in this setting, we applied JOINTLY and scVelo (Bergen et al. (2020): Generalizing RNA velocity to transient cell states through dynamical modeling, Nature Biotechnology) to three batches of cells undergoing erythropoiesis from an atlas of mouse gastrulation (Pijuan-Sala et al. (2019): A single-cell molecular map of mouse gastrulation and early organogenesis, Nature). We found that JOINTLY orders cells along cell types and along latent time in a comparable manner to scVelo (**NEW ANALYSIS** Figure 4I, Supplementary Figure 4E). Interpretation of JOINTLY revealed that the JOINTLY factors have different temporal profiles (**NEW ANALYSIS** Figure 4J). Pathway analysis of genes associated with early, temporary, early-mid, late-mid, or late latent time revealed signalling cascades with different timing, consistent with existing literature on erythropoiesis (**NEW ANALYSIS** Figure 4K).

Point 6:

The application of JOINTLY is another important question. For my understanding, JOINTLY itself does not have a function to do gene expression alignment across conditions. This makes JOINTLY difficult to explore novel cell-type markers or differentially expressed (DE) genes. In this case, JOINTLY can be easily replaced by other mainstream data-integration tools when they find similar cell types. The authors showed JOINTLY application in a white adipose tissue atlas by guiding cell types in data integration using scANVI (Fig. 5), but they did not provide the necessity of JOINTLY in the process. If the authors can show outperformance of JOINTLY + scANVI when compared with other integration tools, the conclusion might be more persuasive.

RESPONSE: JOINTLY does not have an in-built method for gene expression alignment across batches. The main reason for this is that using batch-corrected expression values are generally considered an invalid statistical approach (since integration inherently introduces dependencies between data points, thereby violating its assumptions) and has been shown to have less power than using either a meta-analytical approach or providing the batch variable as a covariate during statistical testing (Nguyen et al. (2023); Benchmarking integration of single-cell differential expression, Nature Communications). However, we recognize that batch-aware imputed expression values can have benefits for data visualization in datasets with large batch effects. To that end, we have added a section on the GitHub repository about how the matrices learned by

JOINTLY can be used for batch-aware imputation of expression values (LINK: <https://github.com/madsen-lab/rJOINTLY>). Briefly, the method involves generating a cell-cell similarity matrix from the learned H_d and F_d matrices $S_d = F_d * H_d$ resulting in a $n_d \times n_d$ matrix. This matrix is made symmetrical by $S_d = S_d * S_d^T$ and used for batch-aware imputation based on the methodology published in Palantir (Setty et al. (2019): Characterization of cell fate probabilities in single-cell data with Palantir, Nature Biotechnology).

We agree with the reviewer that we did not showcase the necessity of JOINTLY in building WATLAS. We do not think that it is in scope to create 8 different atlases of white adipose tissue. First, there is no harmonized ground truth across the included datasets, which means that it is not possible to objectively compare methods. Theoretically, we could reuse the SVM-based strategy to relabel cells across studies, but that requires a tissue atlas, which is what we are trying to create, making the process circular. Second, the WATLAS is not meant as another benchmark of methods, we have already earlier in the paper extensively benchmarked JOINTLY against other methods (in Figures 2 and 3). Instead, the WATLAS is a vignette of what a user could possibly use JOINTLY for and as a community resource. We have added a new sentence in the manuscript (**NEW TEXT** page 6) detailing that WATLAS serves as a vignette of JOINTLY and a community resource.

Minor comments:

(1) The authors may notify that JOINTLY uses clustering results of other cluster methods in the “Introduction” section. Some interpretations, such as “JOINTLY achieves state-of-the-art joint clustering performance, but unlike existing methods with comparable clustering performance, JOINTLY also generates ...”, may mislead the audience that JOINTLY has its own clustering method.

RESPONSE: We agree with the reviewer that the text can be misread and have updated the text (**NEW TEXT** page 2) such that the text is not misleading.

(2) Typo “1.300”, “~10.000”, ...

RESPONSE: We have corrected the thousand separators throughout the manuscript.

(3) Typo “in addition to joint clustering ...”

RESPONSE: We have corrected this, and additional typos through the manuscript.

(4) The audience cannot know the quantitative difference in Fig. 2B and 2C.

RESPONSE: The quantitative values for all metrics used in Figure 2 are available in Supplementary Table S1.

(5) The statement of JOINTLY’s “interpretable” may not be accurate. JOINTLY, LIGER and other NMF-based utilize cell-type-specific gene modules to decompose and integrate datasets, while fastMNN, Scanorama, Seurat, and et al use shared cell types, the cell-type-specific gene modules can also be generated based on cell types after integration.

RESPONSE: JOINTLY is indeed interpretable, as it as the reviewer correctly points out uses gene modules to decompose and integrate datasets. Methods, such as fastMNN, Scanorama and Seurat are not interpretable

because gene modules driving integration cannot be directly extracted. However, we agree with the reviewer that it is possible to extract gene modules post-hoc by performing for example differential expression analysis between clusters (or cell types). The post-hoc method of finding marker genes is poised to identify mostly cluster- or cell type-specific genes, whereas the interpretable methods, including JOINTLY, do not have this limitation, and can discover modules that are shared between cell types. We describe an example of that in Figure 4F. To further understand, the difference between JOINTLY modules and marker genes, we performed a systematic comparison for the adipose tissue dataset. As expected, we found that modules can contain markers from several cell types, but also that module genes are more discriminatory between cell types than marker genes and more conserved across batches. Thus, although certainly possible, the post-hoc analysis of marker genes provides a different view on the data compared to the interpretation of the JOINTLY integration. We have added the following new paragraph (**NEW TEXT** page 6) with several new analyses:

We compared the modules identified by JOINTLY to the list of marker genes for each cell type and found a high overlap between modules and marker genes (**NEW ANALYSIS** Supplementary Figure 4D). However, we found that the marker genes, which are also module genes, have a significantly higher area under the curve (AUC) than marker genes, which are not in the most enriched module (**NEW ANALYSIS** Figure 4D), indicating that module genes are more discriminatory between cell types than marker genes. Similarly, we found that module genes, which are not marker genes, have a significantly lower AUC than genes, which are neither marker nor module genes (**NEW ANALYSIS** Figure 4D) indicating these module genes are markers of other cell types. Finally, we evaluated the consistency of markers across batches by finding marker genes in each batch independently and found that marker genes, which are also module genes, are more often markers in multiple batches compared to marker genes, which are not module genes (**NEW ANALYSIS** Figure 4E). Collectively, this indicates that module genes are highly discriminatory between cell types, and more so than marker genes.

(6) In differential expression analysis of the pseudo-bulk counts, the authors did not show how to correct batch effects across datasets and how to utilize pseudo-bulk analysis to represent transcriptomic characters of single cell counts. Because this is a complicated process³, the authors may provide more details. **RESPONSE:** The use of pseudo-bulk counts is currently the gold-standard for sensitive analysis of transcriptomic differences using single cell RNA-seq datasets (Soneson and Robinson (2018): Bias, robustness and scalability in single-cell differential expression analysis, Nature Methods). To facilitate the reproducibility of this, and all other analyses in the paper, we have released a GitHub repository containing all code necessary to rerun the entire analysis in the manuscript (https://github.com/madsen-lab/JOINTLY_reproducibility).

(7) Fig. 4F and 5H are not in the text.

RESPONSE: We have checked and corrected all missing or wrong figure references.

Reviewer #2

The main topic of this manuscript is the development of an algorithm called JOINTLY for joint clustering of single-cell and single-nucleus RNA-sequencing datasets across batches. The algorithm uses a standardised H matrix as input for clustering and visualisation, and the V matrices are used to identify and score gene modules. The genes are assigned to each factor module by ordering the gene scores within each factor and finding the inflection point of the score distribution using the unit invariant knee method. Finally, the factor modules are scored in each cell using UCell. The paper demonstrates how JOINTLY can be used to create a tissue atlas by clustering and labelling cell types and states in white adipose tissue from 6 different studies. Based on these labels, the authors create a reference atlas of white adipose tissue (WATLAS) deeply characterises the transcriptome of 43 cell types and states. While the paper is clearly written, it would benefit from a more specific description of the key “innovation”. For example, what specific aspect of interpretability or what is the actual gap it is trying to address.

RESPONSE: We thank the reviewer for recognizing the utility of JOINTLY and clearly of the manuscript. Please find below and point-by-point response:

Point 1:

The authors have effectively utilized matrix decomposition as a foundational element in their method. While this approach provides enhanced robustness, it could benefit from further exploration to match the capabilities exhibited by contemporary deep learning-based alternatives. Additionally, enhancing the evaluation process through a comprehensive comparison with modern deep learning-based competitors would enrich the overall analysis.

RESPONSE: There are numerous methods for integrating single-cell and single-nucleus RNA-seq datasets, and new methods are published almost weekly. Here, we compared JOINTLY to the subset of available methods that are widely used in the field and achieve good performance (Luecken et al. (2022): Benchmarking atlas-level data integration in single-cell genomics, Nature Methods), which the reviewer also refers to below. Among those methods is already one deep learning-based alternative, namely scVI, which uses a variational autoencoder to embed samples in a shared latent space.

However, recently, there has been a new development in the field, as the first foundational large language models have been published (scBERT, scFoundation, scGPT). These models represent a paradigm shift, as the underlying concept is that new data is analyzed not from scratch, but by fine-tuning these models. To account for these new developments in the field, we have added scGPT (Cui et al. (2023): scGPT: Towards Building a Foundation Model for Single-Cell Multi-omics Using Generative AI, bioRxiv) to our benchmark and find that scGPT has average performance in terms of joint clustering (**NEW ANALYSIS** Figure 2A-B and Supplementary Figure 2A), but as good robustness against overcorrection as JOINTLY (**NEW ANALYSIS** Figure 3A and Supplementary Figure 3A-B).

Point 2:

While the title mentioned “interpretable clustering,”, the authors have mainly focused on the challenge of batch effect elimination and mitigating the risk of over-correction. Thus the comparison study will be benefit from comparison with data integration approaches mentioned in “Benchmarking atlas-level data integration in single-cell genomics” (<https://www.nature.com/articles/s41592-021-01336-8”>). Following from

point 1, possibly calculating statistics and metrics described in the benchmarking paper so that direct comparison can be made without re-running all the methods.

RESPONSE: We agree with the reviewer that a substantial part of our manuscript deals with batch effect elimination and mitigation of the risk of over-correction. Therefore, and like what is suggested by the reviewer here, we have compared JOINTLY to the best-performing unsupervised methods (Luecken et al. (2022): Benchmarking atlas-level data integration in single-cell genomics, Nature Methods). Among the top 8 methods in that paper, 6 are unsupervised and all 6 are included in our benchmark in Figure 2. Additionally, we have included LIGER, which is another NMF-based interpretable integration method, and now also scGPT (see point 1 above).

For evaluation of our benchmark, we were also inspired by the paper mentioned by the reviewer, and all metrics used in our paper are also used in this benchmarking paper, and they are computed and transformed in similar ways including ARI, cell type and integration average silhouette width and cell type- and integration local inverse Simpson's index making the metrics comparable. However, the approaches are not identical, as we evaluated integration and clustering in R, while sclB is written in Python.

3. In the abstract, the statement, "Unlike existing state-of-the-art methods, JOINTLY is interpretable,..." is an overclaim. Numerous interpretable clustering methods already exist, and many techniques permit post-hoc processing or DE analysis to enhance interpretability. So exactly what is the additional "interpretation" one gains from JOINTLY? In the introduction, the authors referenced LIGER, but there are other clustering methods specifically designed for single-cell data, such as <https://academic.oup.com/bib/article/24/4/bbad199/7176310> that claim to be interpretable. It is vital to succinctly articulate the unique differences and innovations this paper offers compared to the current literature.

RESPONSE: We agree with the reviewer that there are several interpretable methods available, and additional new methods are being published often. We have removed the statement 'Unlike existing state-of-the-art methods' from the abstract (**NEW TEXT** page 1) to tone down the claim that JOINTLY is the only interpretable state-of-the-art method.

If we look at the mainstream methods, which see wide adaptation in the field, we find that they are mostly not directly interpretable, with the notable exception of LIGER. Here, we have benchmarked JOINTLY against these mainstream methods, which achieve state-of-the-art performance in independent benchmarks (Luecken et al. (2022): Benchmarking atlas-level data integration in single-cell genomics, Nature Methods) and find that the integration performance of JOINTLY is on par with the best of the not directly interpretable methods and surpasses that of LIGER.

We also agree with the reviewer that it is possible to define gene modules post-hoc using for example differential expression analysis. To better evaluate the value of interpretable factors, we compared the gene modules derived from the interpretation of JOINTLY to gene modules discovered from differential expression analysis. Generally, we did find a high similarity as measured by the Jaccard index between the interpretable modules and marker gene modules (**NEW ANALYSIS** Supplementary Figure 4D). However, we found that the marker genes, which also are interpretable module genes, have a significantly higher area under the curve (AUC) than marker genes, which are not in the most enriched interpretable module (**NEW ANALYSIS** Figure 4D), indicating that module genes are more discriminatory between cell types than marker genes. Similarly, we found that interpretable module genes, which are not marker genes, have a significantly lower AUC than genes, which are neither marker nor interpretable module genes (**NEW ANALYSIS** Figure 4D) indicating these

interpretable module genes are markers of other cell types. Finally, we evaluated the consistency across batches by finding marker genes in each batch independently, and found that marker genes, which are also interpretable module genes, are more often markers in multiple batches compared to marker genes, which are not interpretable module genes (**NEW ANALYSIS** Figure 4E). In addition, we found that interpretable modules are not limited to finding genes marking a single cell type but can characterize complex patterns shared between subsets of cells across different cell types (see Figure 4F-H). Collectively, this indicates that interpretable module genes are highly discriminatory between cell types, more so than marker genes and that they can reveal biological insights that cannot easily be discovered by conventional differential expression analysis between clusters.

Point 4:

Currently, I believe it's important to recognize that both supervised and unsupervised clustering methods aim to identify "cell types". It's important that in the evaluation the focus is not "narrow" towards "cell type discovery only". One should also consider how JOINTLY performed with methods that incorporate biological knowledge.

RESPONSE: The most common way of seeding models with biological knowledge is to label each sample individually and then use the resulting labels for supervised integration. This approach is taken by for example scANVI (Xu et al. (2021): Probabilistic harmonization and annotation of single-cell transcriptomics data with deep generative models, Molecular Systems Biology) and scGen (Lotfollahi et al. (2019): scGen predicts single-cell perturbation responses, Nature Methods). We know from systematic benchmarks (Luecken et al. (2022): Benchmarking atlas-level data integration in single-cell genomics, Nature Methods) that such supervised integration methods outperform their unsupervised counterparts. In JOINTLY, it is currently not possible to add additional knowledge, and therefore, we believe that comparison to supervised integration methods is out of scope and that their performance is likely to supersede that of all methods tested in the manuscript. We have added a short paragraph about this in the Discussion (**NEW TEXT** page 8). We did, in the last vignette of the paper, on the white adipose tissue atlas, highlight the symbiotic use of JOINTLY and supervised integration using scANVI.

An alternative way of seeding models with biological knowledge is the use of prior knowledge about gene-gene relationships. One example of such a method is scGPT, which has been trained on large amounts of single-cell data to learn gene-gene interactions at the single-cell level. New datasets can be integrated by leveraging these pre-trained gene-gene interactions through transfer learning. As also stated in point 1, we have incorporated scGPT into our manuscript and found average joint clustering performance (**NEW ANALYSIS** Figure 2A-B and Supplementary Figure 2A), but good robustness against over-correction (**NEW ANALYSIS** Figure 3A and Supplementary Figure 3A-B).

Point 5:

Minor: The clustering results that the authors have presented are satisfactory, but still have the potential for further refinement to achieve a cutting-edge advancement in the field. For example, the ARI results shown in Fig2A of JOINTLY and Seurat (common approach) are both 0.9. Is there datasets or situation where JOINTLY is significant improved from Seurat?

RESPONSE: Yes, in most of our tests: In the Pancreas dataset, where JOINTLY achieves a global ARI of 0.920, while Seurat using the RPCA method achieves a global ARI of 0.882, in the Kidney dataset, where JOINTLY

achieves 0.559 and Seurat achieves 0.492 and in the PBMC dataset, JOINTLY achieves 0.728 while Seurat achieves 0.691. All these raw values are available in Supplementary Table S1 and summarized in Figure 2A-B. Additionally, JOINTLY provides robustness against over-correction and interpretable factors (Figure 3A-B and Supplementary Figure 3A-B).

Point 6:

Minor: There are several typos that need to be corrected. For example, at the bottom in page 12, "Minimizing this term encourages the that the same genes contribute the same factors in the clustering matrix across batches."

RESPONSE: We have reviewed the entire manuscript to improve readability and language.

Code and Software Review Data and source code availability and replicability of the results in the paper. The GitHub code has been thoughtfully organised, incorporating reasonable documentation and utilising optimised C++ functions for enhanced efficiency during matrix decomposition.

RESPONSE: We thank the reviewer for carefully looking into our code and recognizing its thoughtful organization, documentation, and optimizations.

Point 1:

The accessibility of the GitHub link provided in the manuscript for reproducibility purposes (i.e. http://www.github.com/madsen-lab/JOINTLY_reproducibility). Unfortunately, attempting to access the link results in a 404 Error, thereby preventing us from fully evaluating the reproducibility of the paper's results.

RESPONSE: We apologize for not making the repository public prior to initial submission. This has now been rectified, such that the reproducibility code is online at the indicated link.

Point 2:

While the well-structured R code and detailed documentation are commendable aspects, the inability to access the private GitHub repository raises significant obstacles in verifying and replicating the research findings.

RESPONSE: All code, including JOINTLY, reproducibility scripts, Single Cell Portal entries and Zenodo entries have been publicly released to facilitate the use, verification, and replication of our research.

Point 3:

The program lacks explicit documentation for all dependencies, specifying compatible versions that users can install seamlessly. In this case, with the "SharedObject" package (we observed the "package 'SharedObject' is not available for BioConductor version '3.17'" error when trying to install rJOINTLY). This scenario might require users to invest considerable time in troubleshooting and resolving version conflicts or, in some cases, necessitate the creation of a separate R environment solely for running the "rJOINTLY" package.

RESPONSE: We have thoroughly tested the installation of JOINTLY across platforms (Unix, macOS and Windows) and across different versions of R. We have updated the package to ensure that installation only proceeds for recent R versions ($\geq 4.1.0$) for which we have had no problems with installing JOINTLY. In addition, we have updated the GitHub repository with more elaborate installation instructions and package dependencies.

Point 4:

Ability to run the tool successfully. Due to insufficient documentation regarding compatible versions of dependencies, package installation was challenging and time consuming. This obstacle made it cumbersome to set up the necessary environment for running the tool. A possible solution is to provide a CodeOcean link with an installed R environment. On the other hand, the test data, i.e., the file `"../data/data_seurat.rds"` mentioned in README is not provided, so we used our own dataset for testing. The main "jointly" function could be successfully executed in our test.

RESPONSE: We are happy to hear that the reviewer successfully ran the main function of JOINTLY. To ensure that users can test JOINTLY, we have now shared all the datasets used for benchmarking JOINTLY on Zenodo (<https://zenodo.org/record/8298157>) and created a walkthrough on GitHub using the human liver dataset.

Point 5:

Code documentation for following the algorithm. The code has been thoughtfully organized and reasonably documented. It is noteworthy that the code utilized C++ scripts, which would be a little bit difficult for users who primarily use R. In addition, a more detailed tutorial will increase usability.

RESPONSE: We have improved the tutorial hosted on the GitHub repository, such that it goes through running JOINTLY, clustering on the resulting embedding, evaluating clusters against labels using ARI and NMI and interpreting JOINTLY to get modules and plotting them. We agree that C++ code can be difficult to read for users with a life science background. However, the bulk of the code is written in R, and only low-level functions pertaining to matrix multiplication are handled using C++. Thus, understanding of the major analytical steps and choices in the algorithm does not necessitate understanding the C++ code. Additionally, the use of C++ code significantly speeds up JOINTLY, and we cannot change this without compromising the runtime of the algorithm.

Point 6:

Software to be run on a widely available operating system. We encountered many challenges in package installation, which was mainly caused by version conflicts of dependencies. Once the conflicts are handled, there would be no reason for the program to be successfully run on a widely available operation system.

RESPONSE: We thank the reviewer for the positive feedback on the quality of the software. We have now tested JOINTLY on several operating systems to ensure compatibility and updated the documentation on dependencies.

Reviewer #1 (Remarks to the Author):

Thanks for the authors' quick response and hard work to revise the manuscript. The additional evidence addresses some of my questions, but the others are still not clear. According to the scores in the supplementary tables, JOINTLY and other existing tools showed similar performance in most benchmark tests, the advantages and outperformance of JOINTLY are not displayed persuasively in the current version. The authors may need to design more scenarios to show the innovation of JOINTLY in data integration analysis and downstream applications.

(1) Major point 2, the authors evaluated the consistency of JOINTLY, but the authors only provided the benchmark results (Fig. 2D ~ 2I) without the integrated data (the output of JOINTLY). This makes the audience difficult to estimate how many cell-types and which ones are affected by this shortcoming. For myself, I agree with the authors' explanation that the performance of most integration tools is "dependent on the number and nature of datasets that are supplied to the algorithm". But whether the major cell types are preserved after integration is important for a statistical tool. The authors may attach the JOINTLY results about the consistency tests.

(2) In the major point 4, the authors' revised work displayed the quantitative values of JOINTLY performance (by the way, the authors may provide row names for all the pages in the supplementary tables, the current version is unreadable).

However, the discussion is still not systematic or unbiased. In the endothelial cells, JOINTLY showed good performance; but the performance of JOINTLY was not good enough when we think about the cell-type relationship among lung ciliated cell, basal cell, and respiratory goblet cell, based on other groups' documented work¹⁻⁴. The distance among these three cell types might be more reasonable in the results of scVI and Harmony.

For myself, the authors may need to provide more scenarios to show the advantages of JOINTLY, which is important for us to agree with this new method.

(3) About WATLAS in my major point 6, if the authors think "there is no harmonized ground truth across the included datasets", then how to judge whether JOINTLY generated correct results, especially, this manuscript does not contain any experiential evidence.

I strongly encourage the authors to show the outperformance and advantages of JOINTLY in the manuscript. If WATLAS is not reliable, the authors can depict different scenario with other datasets.

(4) Minors: in the authors' response point 1, the typo UT * XdT, the shape of UT is (k x m) while that of XdT is (n x m).

Reference:

1. Basil MC, Cardenas-Diaz FL, Kathiriya JJ, Morley MP, Carl J, Brumwell AN, Katzen J, Slovik KJ, Babu A, Zhou S, Kremp MM, McCauley KB, Li S, Planer JD, Hussain SS, Liu X, Windmueller R, Ying Y, Stewart KM, Oyster M, Christie JD, Diamond JM, Engelhardt JF, Cantu E, Rowe SM, Kotton DN, Chapman HA, Morrissey EE. Human distal airways contain a multipotent secretory cell that can regenerate alveoli. *Nature*. 2022 Apr;604(7904):120-126. doi: 10.1038/s41586-022-04552-0. Epub 2022 Mar 30. PMID: 35355013; PMCID: PMC9297319.
2. Kadur Lakshminarasimha Murthy P, Sontake V, Tata A, Kobayashi Y, Macadlo L, Okuda K, Conchola AS, Nakano S, Gregory S, Miller LA, Spence JR, Engelhardt JF, Boucher RC, Rock JR, Randell SH, Tata PR. Human distal lung maps and lineage hierarchies reveal a bipotent progenitor. *Nature*. 2022 Apr;604(7904):111-119. doi: 10.1038/s41586-022-04541-3. Epub 2022 Mar 30. PMID: 35355018; PMCID: PMC9169066.
3. Habermann AC, Gutierrez AJ, Bui LT, Yahn SL, Winters NI, Calvi CL, Peter L, Chung MI, Taylor CJ, Jetter C, Raju L, Roberson J, Ding G, Wood L, Sucre JMS, Richmond BW, Serezani AP, McDonnell WJ, Mallal SB, Bacchetta MJ, Loyd JE, Shaver CM, Ware LB, Bremner R, Walia R, Blackwell TS, Banovich NE, Kropski JA. Single-cell RNA sequencing reveals profibrotic roles of distinct epithelial and mesenchymal lineages in pulmonary fibrosis. *Sci Adv*. 2020 Jul 8;6(28):eaba1972. doi: 10.1126/sciadv.aba1972. PMID: 32832598; PMCID: PMC7439444.
4. Schupp JC, Adams TS, Cosme C Jr, Raredon MSB, Yuan Y, Omote N, Poli S, Chioccioli M, Rose KA, Manning EP, Sauler M, DeIuliis G, Ahangari F, Neumark N, Habermann AC, Gutierrez AJ, Bui LT, Lafyatis R, Pierce RW, Meyer KB, Nawijn MC, Teichmann SA, Banovich NE, Kropski JA, Niklason LE, Pe'er D, Yan X, Homer RJ, Rosas IO, Kaminski N. Integrated Single-Cell Atlas of Endothelial Cells of the Human Lung. *Circulation*. 2021 Jul 27;144(4):286-302. doi: 10.1161/CIRCULATIONAHA.120.052318. Epub 2021 May 25. PMID: 34030460; PMCID:

PMC8300155.

Reviewer #2 (Remarks to the Author):

I thank the authors for their responses and the newly revised manuscript is much improved. While the authors have answered majority of the questions raised in the previous review. I have a few points to raise.

Regarding point 3

- I suggest the statement to be tone down to "JOINTLY is an interpretable state-of the-art method" I think stating "only" is too strong.
- I am confused by Supplementary Figure 4D. The Jaccard index varied between 0 and 0.6, and we see that some factors contain multiple cell types and others have none? Some clarity is needed to show how this figure represents high similarity between gene modules derived from JOINTLY vs DE genes. Could a more direct graphical display (scatter plot be used).
- Could you better clarify again how JOINTLY increase interpretability from DE analysis? I suggest you add a discussion on this component in the manuscript.

Minor comment (point 5): Regarding the analysis based on ARI between JOINTLY and Seurat, the observed ARI difference is limited, is there an estimate of the spread (or confidence) associate with this difference?

Minor comment: Given JOINTLY is performing very similar to large-scale pre-train with an interpretable model, I believe the author have a great opportunity here to state their advantage even stronger in the current manuscript.

About the code

We evaluated the reproducibility of their scripts and the authors have also enhanced their README documents for clarity.

Reviewer #3 (Remarks to the Author):

I co-reviewed this manuscript with one of the reviewers who provided the listed reports as part of the Nature Communications initiative to facilitate training in peer review and appropriate recognition for co-reviewers.

Point-by-point response to reviewers

Reviewer #1:

Thanks for the authors' quick response and hard work to revise the manuscript. The additional evidence addresses some of my questions, but the others are still not clear. According to the scores in the supplementary tables, JOINTLY and other existing tools showed similar performance in most benchmark tests, the advantages and outperformance of JOINTLY are not displayed persuasively in the current version. The authors may need to design more scenarios to show the innovation of JOINTLY in data integration analysis and downstream applications.

RESPONSE: We thank the reviewer for recognizing the hard work that has gone into the revision and that the additional evidence addresses part of the reviewer's concerns. We agree with the reviewer that several methods achieve similar scores in the benchmarking (especially JOINTLY, scVI, Harmony and FastMNN are strong performers and achieve similar global scores for ARI, NMI, cLISI, iLISI, bASW and cASW).

In addition to performing state-of-the-art integration, JOINTLY provides several additional benefits. These benefits include uniform integration, robustness against over-correction, and interpretability.

1. Uniform integration (Figure 2): In the evaluation of the integration of the worst dataset in all benchmarking datasets JOINTLY and scVI are tied for the best rank, while Harmony and FastMNN rank 6th and 4th respectively. This indicates that JOINTLY (and scVI) more effectively integrate all datasets in a sample.
2. Robustness against over-correction (Figure 3). Evaluation of over-correction shows that JOINTLY (and scVI) is robust against over-correction, while for example Harmony and FastMNN are susceptible to cell state erasure. Evaluation of the integration of each tissue in these mixed tasks shows that JOINTLY generally performs better than scVI in terms of integrating both tissues in the mixture.
3. Interpretability (Figure 4): Among the top 4 methods (JOINTLY, scVI, Harmony and FastMNN), only JOINTLY is an intrinsically interpretable method directly identifying active gene modules. We show that these modules can be used to label datasets, are more discriminatory for cell types and more reproducible between batches than traditional marker genes. Finally, we show that these modules also represent transcriptional programs shared between cell types or active in subsets of cells. These modules are difficult to identify in traditional cluster-based differential expression analysis.

Among the tested methods, only JOINTLY, have all these three additional characteristics (uniform integration, robustness against over-correction and interpretability), while also providing state-of-the-art global integration. For those reasons, we strongly believe that JOINTLY represents an important innovation in the field and that it will enable researchers to perform better analyses of their single-cell and single-nucleus RNA-seq datasets.

(1) Major point 2, the authors evaluated the consistency of JOINTLY, but the authors only provided the benchmark results (Fig. 2D ~ 2I) without the integrated data (the output of JOINTLY). This makes the audience difficult to estimate how many cell-types and which ones are affected by this shortcoming. For myself, I agree with the authors' explanation that the performance of most integration tools is "dependent on the number and nature of datasets that are supplied to the algorithm". But whether the major cell types are preserved after integration is important for a statistical tool. The authors may attach the JOINTLY results about the consistency tests.

RESPONSE: We agree that it is important to be able to visually inspect the data. However, due to space constraints, we did not include all 25 UMAPs in the manuscript. We have now included a representative UMAP from each subset, that is one UMAP for a sample with one dataset removed, one for a sample with two datasets removed and one for a sample with three datasets removed (**NEW FIGURE** Supplementary Figure 2B). These plots highlight that the major cell types are well preserved after integration in all subsets. For the reviewer, we have included a panel showing all 25 datasets here:

Figure 1: UMAPs labelled by cell type or clusters for all possible subsets of the human liver dataset with the indicated dataset removed. The cluster labels were chosen to maximize the adjusted Rand index (see Methods in the manuscript).

(2) In the major point 4, the authors' revised work displayed the quantitative values of JOINTLY performance (by the way, the authors may provide row names for all the pages in the supplementary tables, the current version is unreadable). However, the discussion is still not systematic or unbiased. In the endothelial cells, JOINTLY showed good performance; but the performance of JOINTLY was not good enough when we think about the cell-type relationship among lung ciliated cell, basal cell, and respiratory goblet cell, based on other groups' documented work¹⁻⁴. The distance among these three cell types might be more reasonable in the results of scVI and Harmony. For myself, the authors may need to provide more scenarios to show the advantages of JOINTLY, which is important for us to agree with this new method.

RESPONSE: We thank the reviewer for noticing an inadequate separation of lung ciliated cells, basal cells, and respiratory goblet cells in JOINTLY. We have analyzed the issue in detail and found that due to the very complex and heterogeneous batch effects that arise from artificially mixing tissues, the consensus PCA approach was unable to adequately describe the variance in the dataset. For the evaluation of overcorrection, we have rerun the entire analysis using regular PCA, and we now see an appropriate separation of lung ciliated cells, basal cells, and respiratory goblet cells (**NEW FIGURE 3B**), while the endothelial cells from the two tissues remain separated. In terms of metrics, JOINTLY remains on the 2nd overall rank after scGPT. We have added new text to the discussion about when an analyst should choose regular PCA or consensus PCA (**NEW TEXT pp. 8**). In addition to rerunning this analysis, we also expanded evaluation from using only a mixture of the Lung and the Pancreas dataset to all pairwise combinations of datasets (n = 10) used in the benchmarking including both ranks (**NEW FIGURE 3A**) and quantitative metrics (**NEW SUPPLEMENTARY TABLE S2**). Generally, we find that JOINTLY has the best average overall rank across the 10 datasets, and generally achieves a good trade-off between batch correction in each tissue while limiting overcorrection. Finally, we have improved the layout of the supplementary tables to improve readability.

(3) About WATLAS in my major point 6, if the authors think "there is no harmonized ground truth across the included datasets", then how to judge whether JOINTLY generated correct results, especially, this manuscript does not contain any experiential evidence. I strongly encourage the authors to show the outperformance and advantages of JOINTLY in the manuscript. If WATLAS is not reliable, the authors can depict different scenario with other datasets.

RESPONSE: We apologize for the clumsy choice of words. What we meant by that is "there is no harmonized ground truth across the included datasets" is that for the datasets, where author labels are available, the labels are heterogeneous. That is to say that in one study adipocytes may be labelled as 'ADIPO', whereas in another they are labelled as 'Adipocytes', and this label heterogeneity becomes worse for more fine-grained labels. The field has yet to adopt a cell type ontology. To overcome that issue and compare the labels we have defined with author labels, we have now included overlap analysis for each study that has author labels publicly available (**NEW FIGURE Supplementary Figure S5C**), which clearly shows very strong agreement between the labels in WATLAS and labels in the original publication, albeit with different cell type or subtype terminology. This combined with the detailed marker gene analysis for both fine-grained and coarse-grained labels (**Supplementary Figure S5D**) provides strong evidence that the labels defined in WATLAS are of high quality and that WATLAS is a reliable resource for the community.

(4) Minors: in the authors' response point 1, the typo $UT * XdT$, the shape of UT is $(k \times m)$ while that of XdT is $(n \times m)$.

RESPONSE: We thank the reviewer for identifying this typo; X_d is not transposed in the calculation; $U^T (k \times m) * X_d (m \times n)$ yields $E_d (k \times n)$. We have updated the Supplementary Note to correct the typo.

Reference:

1. Basil MC, Cardenas-Diaz FL, Kathiriya JJ, Morley MP, Carl J, Brumwell AN, Katzen J, Slovik KJ, Babu A, Zhou S, Kremp MM, McCauley KB, Li S, Planer JD, Hussain SS, Liu X, Windmueller R, Ying Y, Stewart KM, Oyster M, Christie JD, Diamond JM, Engelhardt JF, Cantu E, Rowe SM, Kotton DN, Chapman HA, Morrissey EE. Human distal airways contain a multipotent secretory cell that can regenerate alveoli. *Nature*. 2022 Apr;604(7904):120-126. doi: 10.1038/s41586-022-04552-0. Epub 2022 Mar 30. PMID: 35355013; PMCID: PMC9297319.
2. Kadur Lakshminarasimha Murthy P, Sontake V, Tata A, Kobayashi Y, Macadlo L, Okuda K, Conchola AS, Nakano S, Gregory S, Miller LA, Spence JR, Engelhardt JF, Boucher RC, Rock JR, Randell SH, Tata PR. Human distal lung maps and lineage hierarchies reveal a bipotent progenitor. *Nature*. 2022 Apr;604(7904):111-119. doi: 10.1038/s41586-022-04541-3. Epub 2022 Mar 30. PMID: 35355018; PMCID: PMC9169066.
3. Habermann AC, Gutierrez AJ, Bui LT, Yahn SL, Winters NI, Calvi CL, Peter L, Chung MI, Taylor CJ, Jetter C, Raju L, Roberson J, Ding G, Wood L, Sucre JMS, Richmond BW, Serezani AP, McDonnell WJ, Mallal SB, Bacchetta MJ, Loyd JE, Shaver CM, Ware LB, Bremner R, Walia R, Blackwell TS, Banovich NE, Kropski JA. Single-cell RNA sequencing reveals profibrotic roles of distinct epithelial and mesenchymal lineages in pulmonary fibrosis. *Sci Adv*. 2020 Jul 8;6(28):eaba1972. doi: 10.1126/sciadv.aba1972. PMID: 32832598; PMCID: PMC7439444.
4. Schupp JC, Adams TS, Cosme C Jr, Raredon MSB, Yuan Y, Omote N, Poli S, Chioccioli M, Rose KA, Manning EP, Sauler M, Deluliis G, Ahangari F, Neumark N, Habermann AC, Gutierrez AJ, Bui LT, Lafyatis R, Pierce RW, Meyer KB, Nawijn MC, Teichmann SA, Banovich NE, Kropski JA, Niklason LE, Pe'er D, Yan X, Homer RJ, Rosas IO, Kaminski N. Integrated Single-Cell Atlas of Endothelial Cells of the Human Lung. *Circulation*. 2021 Jul 27;144(4):286-302. doi: 10.1161/CIRCULATIONAHA.120.052318. Epub 2021 May 25. PMID: 34030460; PMCID: PMC8300155.

Reviewer #2:

I thank the authors for their responses and the newly revised manuscript is much improved. While the authors have answered majority of the questions raised in the previous review. I have a few points to raise.

Regarding point 3

- I suggest the statement to be tone down to “JOINTLY is an interpretable state-of the-art method” I think stating “only” is too strong.

RESPONSE: We have toned down any statements about JOINTLY being the only state-of-the-art interpretable method.

- I am confused by Supplementary Figure 4D. The Jaccard index varied between 0 and 0.6, and we see that some factors contain multiple cell types and others have none? Some clarity is needed to show how this figure represents high similarity between gene modules derived from JOINTLY vs DE genes. Could a more direct graphical display (scatter plot be used).

RESPONSE: This observation that factors are diverse in enrichment in DE genes derived from cell types highlights the advantages and complementary benefits of interpreting JOINTLY relative to regular cluster-based differential expression analysis. Namely, the interpretable factors are not limited to being enriched in a cluster or cell type but can be shared between several cell types that share transcriptional programs or be active only in certain samples or conditions. This allows the analyst to discover active biological programs that would have been missed by cluster-based DE analysis. In the manuscript, we show an example of this in Figure 4F-H, which shows one factor, which is enriched in a subpopulation of FAPs and adipocytes and is associated with insulin sensitivity/signaling.

In the current version of JOINTLY, genes are assigned to an interpretable factor in a binary sense (i.e., yes, or no), and therefore, it is not straightforward to use for example a scatter plot to directly compare log₂ fold changes with the interpretation modules. However, it is an interesting idea to see if it possible to derive gene weights from the solved *W* matrices and use them to better refine module scoring for example. This is out-of-scope of the current manuscript, but we will explore the idea in future work.

Since we do not have a continuous measure for the interpretable factors (to compare to for example log₂ fold changes in traditional DE analysis), we used the Jaccard index to measure the overlap between the two lists (marker gene; yes, or no and interpretable factor; yes, or no) while controlling for the size of the lists, such that a Jaccard index of 1 is perfect overlap and 0 is no overlap. We chose to use this metric because it can be easily summarized, visualized, and compared across all 195 comparisons (15 factors by 13 cell types). It is important that we make all 195 comparisons because one factor does not necessarily have to correspond one-to-one to one cell type.

Referring to Supplementary Figure 4D, we find that some factors are very cell type discriminatory (for example factor 6 contains almost exclusively FAP marker genes), whereas others are shared between related cell types (for example factor 10 contains genes marking both FAPs and adipocytes), and as correctly pointed by the reviewer one is not enriched in any cell type (factor 9). This is an interesting factor, as it correlates strongly with the fraction of spliced reads (see figure 2A below) and has a distribution in the extremes/tips of clusters (see figure 2B below).

Figure 2: Analysis of factor 9. **A)** Scatterplot showing the relationship between the score of factor 9 and the percent of reads from exons (spliced transcripts). **B)** UMAP showing the score of factor 9. Arrows indicate cells with higher scores.

This could suggest that this factor is capturing the ambient RNA profile, indicating inadequate data cleanup by the original authors. Disregarding this factor improves the ARI between clusters and author labels from 0.935 to 0.944. This nicely showcases the power of interpretable factors, which can aid analysts in understanding what is driving data integration and use those insights to adjust their analysis.

- Could you better clarify again how JOINTLY increase interpretability from DE analysis? I suggest you add a discussion on this component in the manuscript.

RESPONSE: Following up on the answer to the question above, the main advantages of JOINTLY as compared to DE analysis are twofold:

1. The genes assigned to factors are generally more discriminative and more conserved between batches than traditional DE genes (see Figure 4D).

JOINTLY can discover transcriptional programs that are active in only a subset of cells in a cluster or shared between clusters. These signals are easy to miss in traditional DE analysis. For example, module 10, which is enriched in a subpopulation of FAPs and adipocytes (see Figure 4F). One example of a gene belonging to module 10 is CLSTN2, which is not significantly enriched in FAPs through traditional DE analysis (FDR-adjusted P-value = 0.457, log₂ fold change = -0.00078), but it has a weak but distinct spatial distribution in the embedded space in FAPs (and in adipocytes, where it is a marker of one of the subpopulations).

Figure 3: Expression of CLSTN2 in the Emont et al. dataset.

We have added to the discussion to clarify how interpretability can be used and how it differs from DE analysis (**NEW TEXT** pp. 9).

Minor comment (point 5): Regarding the analysis based on ARI between JOINTLY and Seurat, the observed ARI difference is limited, is there an estimate of the spread (or confidence) associate with this difference?

RESPONSE: Within each dataset, we cannot calculate spread or confidence because Seurat is by default deterministic and produces the same result every time. What we can do instead is to calculate the average ARI across the 5 datasets used for benchmarking. Here, the average ARI for JOINTLY is 0.79, and 0.76 for Seurat. To interpret the difference, it is important to remember that the scale for ARI is between 0 and 1. To put the difference into perspective, in a dataset with 10.000 cells, a decrease of 0.03 in ARI corresponds to approximate 40 additional misclassifications. Thus, in our opinion an increase of 0.03 represents a meaningful increase in classification precision. In addition to the increase in ARI, JOINTLY also provides several additional advantages, such as protection against over-correction (something Seurat does poorly at according to our metrics) and direct interpretability.

Minor comment: Given JOINTLY is performing very similar to large-scale pre-train with an interpretable model, I believe the author have a great opportunity here to state their advantage even stronger in the current manuscript.

RESPONSE: We thank the reviewer for recognizing the advantages of JOINTLY. We have briefly commented on the comparison between scGPT and task-specific models in the discussion (**NEW TEXT** pp. 8).

About the code. We evaluated the reproducibility of their scripts and the authors have also enhanced their README documents for clarity.

RESPONSE: We thank the reviewer for recognizing that we have improved the online code.

Reviewer #1 (Remarks to the Author):

The new revised manuscript and response letter address most of my concerns, thank the authors' detailed explanations. But I want to remind that the core of non-negative matrix factorization (NMF) is to decompose the target matrix (gene-cell matrix) to generate factor matrix, for my understanding, which is "interpretable" part of JOINTLY. The innovation of JOINTLY might be weakened when using this part as evidence. In my mind, based on the method framework, the consensus PCA of JOINTLY may restrictively reduce the bias of data integration with homogenous cell-types, i.e., in data integration, even some (few) datasets with very poor quality, the consensus PCA of JOINTLY might complement them by other good-quality ones. This method might have more advantages in under-correction than in over-correction data integration. I think the authors also recognize this part, "This choice is appropriate for datasets, where the major axes of variation are assumed to be similar, such as biological replicates."

Reviewer #2 (Remarks to the Author):

I thank the authors for their further responses, and I have not further comments on the manuscript. Just a comment for the future: I appreciate that some clustering approach is deterministic, and typically one would consider multiple resampling the original data followed by clusterig to get a sense of the variability surrounding the ARI estimate.

Reviewer #3 (Remarks to the Author):

I co-reviewed this manuscript with one of the reviewers who provided the listed reports as part of the Nature Communications initiative to facilitate training in peer review and appropriate recognition for co-reviewers.

Point-by-point response to reviewers

We thank the reviewers for helpful comments on the manuscript throughout revision, which we believe have helped improve the quality of the manuscript.

Reviewer #1 (Remarks to the Author):

The new revised manuscript and response letter address most of my concerns, thank the authors' detailed explanations. But I want to remind that the core of non-negative matrix factorization (NMF) is to decompose the target matrix (gene-cell matrix) to generate factor matrix, for my understanding, which is "interpretable" part of JOINTLY. The innovation of JOINTLY might be weakened when using this part as evidence. In my mind, based on the method framework, the consensus PCA of JOINTLY may restrictively reduce the bias of data integration with homogenous cell-types, i.e., in data integration, even some (few) datasets with very poor quality, the consensus PCA of JOINTLY might complement them by other good-quality ones. This method might have more advantages in under-correction than in over-correction data integration. I think the authors also recognize this part, "This choice is appropriate for datasets, where the major axes of variation are assumed to be similar, such as biological replicates."

We thank the reviewer for the comments and are happy that we could address the concerns. We agree that consensus PCA is mostly appropriate for homogenous datasets. To integrate heterogeneous datasets with large components of unshared variation, users should apply regular PCA (as for example done for the integration of tissue mixtures).

Reviewer #2 (Remarks to the Author):

I thank the authors for their further responses, and I have no further comments on the manuscript. Just a comment for the future: I appreciate that some clustering approach is deterministic, and typically one would consider multiple resampling the original data followed by clustering to get a sense of the variability surrounding the ARI estimate.

We thank the reviewer for their feedback on our manuscript and agree that for future benchmarks subsampling or using different seeds might be a good strategy. For the present manuscript, we opted to run all alternative methods on the full dataset using default parameters to most closely recapitulate how an end-user might use the algorithms.

Reviewer #3 (Remarks to the Author):

I co-reviewed this manuscript with one of the reviewers who provided the listed reports as part of the Nature Communications initiative to facilitate training in peer review and appropriate recognition for co-reviewers.

We thank the reviewer for their feedback on our manuscript.